# TD3B: Transition-Directed Discrete Diffusion for Allosteric Binder Generation

**Hanqun Cao** [* 1]  **Aastha Pal** [* 2]  **Sophia Tang** [3]  **Yinuo Zhang** [3 4]  **Jingjie Zhang** [1]  **Pheng Ann Heng** [1]
**Pranam Chatterjee** [2 3]

## Abstract

Protein function is often controlled by ligands that bias the direction of state transitions, such as agonists and antagonists, rather than stabilizing a single conformation. This is especially important for clinically relevant G protein-coupled receptors (GPCRs), where therapeutic efficacy depends on functional directionality. Structure-based design methods optimize binding to static conformations and cannot represent non-reversible, directional effects or systematically distinguish agonist from antagonist behavior. To address this gap, we introduce **T**ransition-**D**irected **D**iscrete **D**iffusion for allosteric **B**inder design (**TD3B**), a sequence-based generative framework that designs binders with specified agonist or antagonist behavior via a directional transition control objective. TD3B combines a target-aware Direction Oracle, a soft binding-affinity gate, and amortized fine-tuning of a pre-trained discrete diffusion model, enabling targeted agonist and antagonist generation decoupled from binding affinity and unattainable by equilibrium-based or inference-only guidance baselines. The code and checkpoints are available at https://huggingface.co/ChatterjeeLab/TD3B.

## 1 Introduction

Protein allostery governs regulation and control across signaling, transport, and transcriptional systems, yet its functional effects are fundamentally dynamic (Motlagh et al.,

[1]Department of Computer Science and Engineering, The Chinese University of Hong Kong [2]Department of Bioengineering, University of Pennsylvania [3]Department of Computer and Information Science, University of Pennsylvania [4]Centre for Computational Biology, Duke-NUS Medical School. Correspondence to: Pranam Chatterjee <pranam@seas.upenn.edu>.

*Proceedings of the 43rd International Conference on Machine Learning*, Seoul, South Korea. PMLR 306, 2026. Copyright 2026 by the author(s).

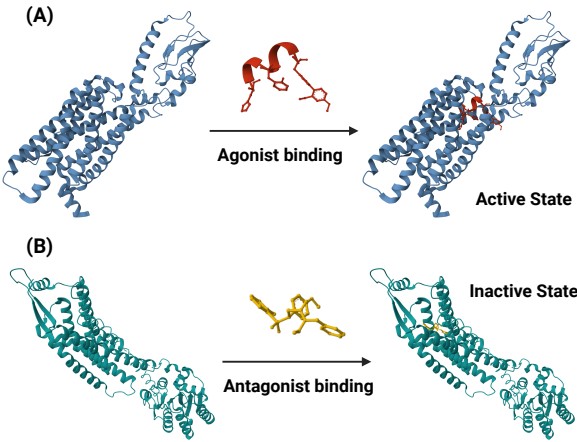

*Figure 1.* Structural mechanisms of agonist and antagonist peptide binding. **(A)** Agonist binding (red) triggers conformational shifts in the target protein (blue) to its Active State, initiating downstream signaling. **(B)** Antagonist binding (yellow) stabilizes the Inactive State by occupying the binding pocket without inducing structural changes, effectively blocking signal activation.

2014; Weikl & Paul, 2014). In many biologically relevant settings, most notably for agonists and antagonists, function depends not on stabilizing a single conformation, but on biasing the direction of transitions between macrostates such as activation and deactivation (Henzler-Wildman & Kern, 2007; Cao et al., 2021). Despite this, most contemporary binder design algorithms, such as RFdiffusion (Watson et al., 2023), BindCraft (Pacesa et al., 2025), BoltzGen (Stark et al., 2025), and GPCR-specialized extensions like RareFoldGPCR (Li et al., 2025), treat proteins as fixed objects and frame design around stabilizing a target structure or interface. While these methods can generate binders or even active agonists, they remain intrinsically tied to equilibrium structural priors and lack a mechanism to represent or control transition asymmetry. As a result, they cannot distinguish or systematically design agonist versus antagonist behavior, since static structures alone do not encode non-reversible, directional effects. Consequently, binders whose function arises from reshaping kinetic pathways rather than stabilizing endpoints lie outside the representational scope of structure-centric design approaches.

Recent advances in discrete generative modeling have produced high-capacity peptide language models that capture the structure of peptide sequence space independent of downstream objectives (Tang et al., 2025a;c; Chen et al., 2025b;a; Vincoff et al., 2025; Tang et al., 2025b). In particular, masked discrete diffusion language models (MDLMs) such as PepTune learn strong unconditional priors over valid peptide sequences, decoupling sequence syntax and diversity from task-specific optimization (Tang et al., 2025a;c). Building on this foundation, guidance strategies such as PepTune and TR2-D2 introduce lightweight fine-tuning and objective-guided sampling mechanisms that bias generation toward desired properties without retraining the base model from scratch (Tang et al., 2025a;c). Our approach leverages this separation by treating directional allosteric control as a guidance objective layered on top of an existing discrete diffusion backbone, rather than as a new generative architecture. The central question we address is:

*What form should such a guidance objective take when the desired effect is directional modulation of protein states rather than equilibrium binding?*

In this work, we introduce **T**ransition-**D**irected **D**iscrete **D**iffusion for allosteric **B**inder design (**TD3B**), a discrete generative framework that treats directional allostery as a key design objective. We model binder action through *sequence-conditioned transition operators* over protein macrostates, explicitly accommodating non-reversible, antisymmetric state changes. Our approach leverages directional supervision to bias generation toward binders that promote or suppress specific state transitions. This reframes binder design as a non-equilibrium transport problem, enabling the generation of binders that control protein function by reshaping transition directionality rather than stabilizing conformations.

Our contributions are threefold:

1. **A transition-operator formulation of directional allosteric control.** We formalize binder-mediated allostery as a sequence-conditioned transition operator over protein macrostates, with binary $\pm 1$ supervision capturing the *sign* of transition asymmetry. This makes directionality and non-reversibility explicit modeling targets beyond static or equilibrium-based representations, while remaining honest about the granularity of supervision: we do not regress continuous kinetic rates, but rather use the operator formalism to motivate a coarse, learnable design objective.

2. **A directionally guided generative framework for binder design.** We co-design three components for direction-controlled generation: (i) a target-aware Direction Oracle that classifies agonist vs. antagonist

effect for arbitrary target proteins; (ii) a *gated* reward in which a pre-trained affinity predictor acts as a soft binding gate (rather than as a Pareto objective traded off against direction), and the oracle supplies the directional signal; and (iii) tree-search amortized fine-tuning of a pre-trained MDLM via importance-weighted denoising and a contrastive loss enforcing directional separation in representation space.

3. **Empirical directional selectivity beyond static baselines.** We demonstrate that contrastive, direction-based fine-tuning produces binders that selectively bias agonistic transitions while minimally affecting reverse transitions, capturing functional behaviors that structure-based and inference-only baselines cannot achieve through post-hoc filtering.

## 2 Related Works

### 2.1 Allosteric Binder Design

Classical allosteric theory originated from the MWC concerted transition (Monod et al., 1965) and KNF sequential induced-fit (Koshland Jr et al., 1966) models, while early discovery relied on serendipitous screening hits (Lu et al., 2019). With the expansion of structural databases (Shen et al., 2016), computational methods emerged to predict allosteric sites (Huang et al., 2013; Akbar & Helms, 2018), hotspots (Clarke et al., 2016), and communication pathways (Tan et al., 2019; Halabi et al., 2009) from static structures. However, static approaches miss cryptic sites that remain occluded in certain conformations. Dynamics-based methods using MD simulations and Markov state models (Shukla et al., 2014; Bowman et al., 2015) address this limitation but incur high computational costs.

*De novo* allosteric design has evolved from early Rosetta-based side-chain networks (Churchfield et al., 2016; Pirro et al., 2020) to recent modular rigid-body coupling strategies leveraging RFDiffusion (Watson et al., 2023) and Protein-MPNN (Dauparas et al., 2022), enabling peptide-responsive ring architectures and effector-induced cage disassembly (Pillai et al., 2024). Complementary approaches include structure-based design for GPCRs (Li et al., 2025) and Chemical Language Model (CLM)-based generation (Ballarotto et al., 2023). In contrast, TD3B reframes directional design as a non-equilibrium transport problem, introducing directional supervision such that binding agents act as one-way valves controlling state transition directionality.

### 2.2 Conditional Generation for Discrete Diffusion

Diffusion models have emerged as powerful unsupervised generative frameworks capable of capturing distributions over discrete spaces, with conditional generation enabling efficient sample exploration (Austin et al., 2021; Lou et al.,

2024; Sahoo et al., 2024; Shi et al., 2024). To steer generation toward desired properties, guided generation approaches incorporate classifier gradients into the sampling process. These include training-free methods such as Classifier Guidance (CG) and Sequential Monte Carlo (SMC) (Dhariwal & Nichol, 2021; Nisonoff et al., 2024; Chung et al., 2022; Wu et al., 2023; Dou & Song, 2024; Phillips et al., 2024), as well as Classifier-Free Guidance (CFG) (Ho & Salimans, 2022). While computationally efficient, these approaches struggle to provide complex guidance in discrete domains, where gradient-based steering is inherently limited by non-differentiable operations.

Reinforcement learning-based methods address this limitation by enabling discrete diffusion models to learn more sophisticated conditional distributions through environment interaction. Policy gradient approaches such as DRAKES and GLID$^2$E (Wang et al., 2025; Cao et al., 2025) update model policies via reward signals, while tree-search methods including PepTune and TR2-D2 (Tang et al., 2025a;c) offer greater flexibility for multi-condition and multi-objective sampling by explicitly exploring the combinatorial space.

## 3 Preliminaries

This section reviews the generative modeling primitives underlying our approach. We describe masked discrete diffusion language models (MDLMs) as a general-purpose backbone for sequence generation (Sahoo et al., 2024; Austin et al., 2021; Shi et al., 2024) and introduce amortized objective-guided fine-tuning as a mechanism for incorporating external objectives (Tang et al., 2025c). These components are standard or well-established; we defer the dynamical formulation underlying our method (macrostates, transition operators, and directional asymmetry) and the specifics of our proposed framework to Section 4.

### 3.1 Discrete Sequence Spaces

Let $\mathcal{A}$ denote a finite alphabet and $\mathcal{A}^L$ the space of length-$L$ sequences. In this work, sequences correspond to binders, and the formulation applies to arbitrary sequence-based binders, with peptides serving as a concrete instantiation. We write $y \in \mathcal{A}^L$ for a clean sequence and use $x$ to denote target protein sequences.

### 3.2 Masked Discrete Diffusion Language Models

Masked discrete diffusion language models (MDLMs) define generative processes over discrete sequences by progressively denoising corrupted inputs. A forward noising process $q_t(y_t \mid y)$ maps a clean sequence $x$ to a partially corrupted version $y_t$ at time $t \in [0, 1]$, typically by masking tokens according to a time-dependent schedule. The reverse process is parameterized by a neural network that predicts

the distribution of masked tokens conditioned on the unmasked context and time. Training proceeds by minimizing a denoising cross-entropy objective:

$$
\begin{aligned}
\mathcal{L}_{\mathrm{DCE}}(\theta, y) =& \mathbb{E}_{t \sim \mathcal{U}(0,1)} \left[ \frac{1}{t} \mathbb{E}_{q_t(\tilde{y}_t | y)} \right. \\
& \left. \sum_{\ell: y_t^\ell = \boldsymbol{M}} -\log p_\theta \left( y^\ell \mid y_t^{\mathrm{UM}}, t \right)_{y^\ell} \right],
\end{aligned}
\tag{1}
$$

where $\boldsymbol{M}$ denotes the special mask token, $y_t^{\mathrm{UM}}$ represents the set of unmasked tokens at diffusion time $t$, the index $\ell$ ranges over all masked sequence positions, and $p_\theta(y^\ell \mid y_t^{\mathrm{UM}}, t)$ is the model's predicted distribution over the alphabet $\mathcal{A}$ at position $\ell$. Once trained, an MDLM defines an unconditional distribution over valid sequences and can be sampled by iteratively denoising from a fully masked input. Importantly, the MDLM captures the combinatorial structure of sequence space independently of any downstream task.

### 3.3 Amortized Objective-Guided Fine-Tuning of MDLMs

Let $p_{\theta_0}(y)$ denote a pre-trained MDLM defining an unconditional distribution over sequences. Objective-guided sequence design seeks to bias this distribution toward sequences that score highly under an external objective $S(y)$, while preserving the structural prior learned by $p_{\theta_0}$.

Amortized fine-tuning methods, such as TR2-D2 (Tang et al., 2025c), achieve this by learning a new parameterization $p_\theta(y)$ that approximates a reward-tilted target distribution:

$$
p^\star(x) \propto p_{\theta_0}(x) \exp \left( \frac{S(y)}{\alpha} \right),
\tag{2}
$$

where $\alpha > 0$ is a regularization coefficient controlling the strength of deviation from the pre-trained prior $p_{\theta_0}$, and $S(y)$ is the external scalar objective determining high-scoring sequences that we want the fine-tuned model to up-weight. This tilted distribution is the optimal solution to the KL-regularized objective $\max_p \mathbb{E}_p[S(y)] - \alpha \mathrm{KL}(p \| p_{\theta_0})$: sequences with higher scores are exponentially favored, while $\alpha$ governs the trade-off between staying close to $p_{\theta_0}$ (large $\alpha$) and aggressively concentrating mass on top scorers (small $\alpha$). This formulation defines an implicit energy-based reweighting of the base model without modifying the corruption process or denoising schedule.

Since direct sampling from $p^\star$ is intractable, amortized fine-tuning optimizes $\theta$ via a weighted denoising cross-entropy (WDCE) objective. Let $y_{0:1}$ denote a denoising trajectory

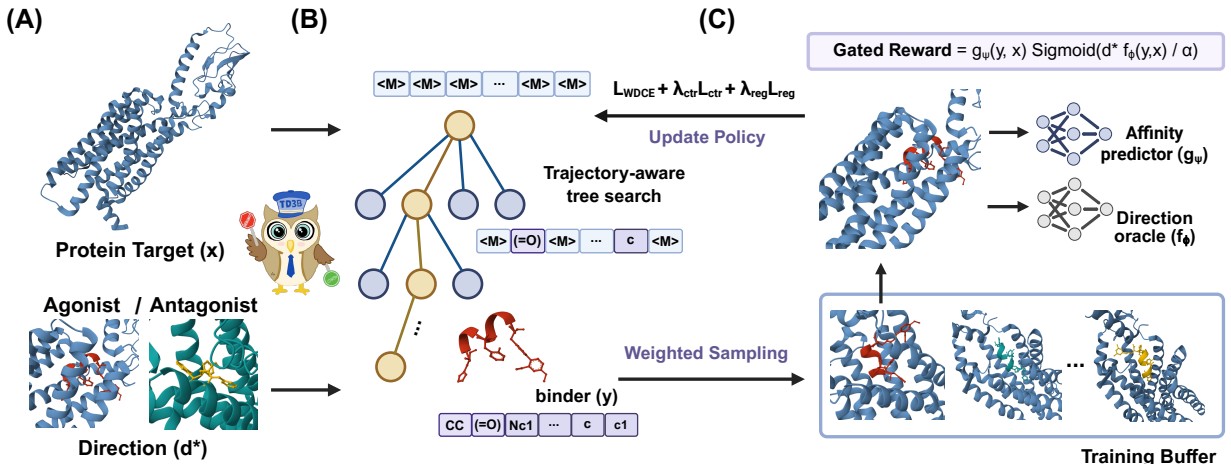

*Figure 2.* **Overview of the TD3B framework.** **(A)** TD3B enables flexible peptide binder design with specified directionality for diverse protein targets; the 3D rendering illustrates the biological setting (active vs. inactive macrostates) — TD3B itself operates on *sequence* representations of both the target protein $x$ and the binder $y$, not on 3D structures. **(B)** Sampling phase: TD3B performs trajectory-aware tree search over masked discrete diffusion trajectories, conditioned on the target sequence and desired direction $d^\star$, and produces binder candidates via weighted sampling. **(C)** Fine-tuning phase: the policy model is updated on samples drawn from a replay buffer, guided by the gated reward (Eq. 21), with combined WDCE, contrastive, and KL terms.

with final sequence $y = y_1$. The fine-tuning objective is:

$$\mathcal{L}_{\text{WDCE}}(\theta) = \mathbb{E}_{p^\star(y)}[\mathcal{L}_{\text{DCE}}(\theta; y)] \tag{3}$$

$$= \mathbb{E}_{y_{0:1} \sim \mathbb{P}^\star}[\mathcal{L}_{\text{DCE}}(\theta; y_1)] \tag{4}$$

$$= \mathbb{E}_{y_{0:1} \sim \mathbb{P}^v}\left[\underbrace{\frac{d\mathbb{P}^\star}{d\mathbb{P}^v}(y_{0:1})}_{w(y_{0:1})} \mathcal{L}_{\text{DCE}}(\theta; y_1)\right], \tag{5}$$

where $\mathbb{P}^\star$ is the path distribution induced by the target $p^\star$, $\mathbb{P}^v$ is a path distribution generated by the current policy from which we can sample from, and $w(y_{0:1})$ is the importance weight that corrects the mismatch between the two. Under the assumption that the trajectory distribution factorizes and the reward depends only on the final sequence, the trajectory-level importance weight decomposes as:

$$w(y_{0:1}) \propto \exp\left(\frac{S(y_1)}{\alpha}\right) \cdot \prod_{n=1}^{T} \frac{p_{\theta_0}(y_{t_{n-1}} \mid y_{t_n})}{p_{\bar{\theta}}(y_{t_{n-1}} \mid y_{t_n})}, \tag{6}$$

where the trajectory is discretized as $0 = t_0 < t_1 < \cdots < t_T = 1$, the first factor is the reward tilt evaluated at the clean sample $y_1$, and the product is an importance ratio correcting for sampling under the proposal $p_{\bar{\theta}}$ rather than the pre-trained reverse process $p_{\theta_0}$.

This procedure internalizes objective $S$ into the model's sampling distribution, minimizing the need for extensive inference-time search. While lightweight methods like best-of-$K$ selection can still refine quality, this amortized initialization is notably more efficient than guidance-only approaches. Fine-tuning updates only the denoising conditionals, leaving the diffusion and corruption schedules intact.

## 4 Problem Formulation

We formalize directional allosteric binder design as an amortized objective-guided sequence generation problem, where a pre-trained MDLM is fine-tuned to generate sequences biasing protein state transitions in a specified direction. At a high level, TD3B proceeds in three stages: **(1)** a target-aware Direction Oracle (Section 4.5) is trained to predict whether a candidate binder promotes activation (agonist) or stabilizes the inactive state (antagonist) given a target protein. **(2)** This oracle is combined with a pre-trained binding-affinity predictor into a *gated reward* (Section 4.7): the affinity model acts as a soft gate that filters non-binders, while the oracle provides the directional signal. **(3)** The gated reward is used to fine-tune a pre-trained discrete diffusion generator via importance-weighted denoising (Section 4.8), with a contrastive loss (Section 4.6) maintaining separation between agonist and antagonist representations and a KL term keeping the fine-tuned policy close to the pre-trained prior (Figure 2). To set up the design objective, we first introduce a coarse-grained dynamical view of protein state transitions (Sections 4.1–4.3), then describe directional supervision and the components of our generative framework (Sections 4.4–4.9).

### 4.1 Protein Macrostates and Coarse-Grained State Shifts

To reason about allosteric function, we adopt a coarse-grained description of protein state shifts grounded in the macrostate / Markov state model literature (Shukla et al.,

2014; Bowman et al., 2015; Noé et al., 2009). Let

$$\mathcal{S} = \{s_1, \ldots, s_K\} \tag{7}$$

denote a finite set of macrostates corresponding to functionally distinct configurations, such as inactive/active or closed/open. In the absence of a binder, protein state shifts are modeled as a continuous-time Markov chain (CTMC) with generator:

$$Q_0 : \mathcal{S} \times \mathcal{S} \to \mathbb{R}, \quad Q_0(s_i, s_i) = -\sum_{j \neq i} Q_0(s_i, s_j). \tag{8}$$

This abstraction captures state-to-state transition behavior without committing to atomistic trajectories or detailed kinetic models.

## 4.2 Sequence-Conditioned Transition Operators

A binder sequence $y \in \mathcal{A}^L$ may alter the transition structure of proteins, consistent with the ligand-induced dynamics view of allostery (Cao et al., 2021; Motlagh et al., 2014). We represent this effect by a sequence-conditioned generator:

$$Q^{(y)} = Q_0 + \Delta Q^{(y)}, \tag{9}$$

where $\Delta Q^{(y)}$ denotes a sequence-dependent perturbation of transition rates. No symmetry or reversibility is assumed. In general, the binder-conditioned generator is asymmetric,

$$Q^{(y)}(s_i, s_j) \neq Q^{(y)}(s_j, s_i), \tag{10}$$

and the resulting dynamics need not satisfy detailed balance:

$$\pi^{(y)}(s_i)Q^{(y)}(s_i, s_j) \neq \pi^{(y)}(s_j)Q^{(y)}(s_j, s_i), \tag{11}$$

While the stationary distribution $\pi^{(y)}$ exists for finite irreducible CTMCs, it does not imply reversibility; thus, these generators lack a scalar energy gradient representation.

## 4.3 Directional Asymmetry

For any ordered state pair $(s_i, s_j)$, we define the directional asymmetry induced by a sequence $y$ as:

$$\Delta_{ij}(y) := Q^{(y)}(s_i, s_j) - Q^{(y)}(s_j, s_i). \tag{12}$$

This quantity captures the net bias of transitions between macrostates. Directional allosteric effects correspond to consistent signs of $\Delta_{ij}(y)$ for selected transitions. We emphasize that neither the absolute magnitude of $\Delta_{ij}(y)$ nor the full generator $Q^{(y)}$ is assumed to be observable; only coarse directional information may be available in practice. The transition-operator formalism above provides motivation and theoretical grounding for our design objective: it explains *why* directionality is fundamentally distinct from static binding (because agonist/antagonist effects arise from asymmetric, non-reversible perturbations of transition rates), but the operator $Q^{(y)}$ itself is not parameterized or regressed during training. Our supervision instead captures only the *sign* of $\Delta_{ij}(y)$, as detailed next.

## 4.4 Data and Directional Supervision

We assume access to a dataset:

$$\mathcal{D} = \{(x^{(n)}, y^{(n)}, a^{(n)})\}_{n=1}^{N}, \tag{13}$$

where $x^{(n)}$ is the target protein sequence, $y^{(n)} \in \mathcal{A}^L$ is a binder sequence, and

$$a^{(n)} \in \{\text{full agonist, partial agonist, antagonist, negative}\}$$

is a categorical functional action label. Negative labels indicate lack of binding and are excluded from directional supervision.

For each target protein, we adopt a two-macrostate abstraction:

$$\mathcal{S} = \{s_{\text{inactive}}, s_{\text{active}}\}. \tag{14}$$

A binder sequence $y$ induces a sequence-conditioned generator $Q^{(y)}$ on $\mathcal{S}$, but neither $Q^{(y)}$ nor its transition rates are observed. Instead, functional labels specify the sign of the induced transition asymmetry:

$$\Delta(y) := Q^{(y)}(s_{\text{inactive}}, s_{\text{active}}) - Q^{(y)}(s_{\text{active}}, s_{\text{inactive}}).$$

We encode supervision using a direction label $d(y) \in \{+1, -1\}$ and a confidence weight $\kappa(y) \in [0, 1]$:

$$d(y) = \begin{cases} +1, & a(y) \in \{\text{full agonist, partial agonist}\}, \\ -1, & a(y) = \text{antagonist}, \end{cases}$$

$$\kappa(y) = \begin{cases} 1, & a(y) = \text{full agonist}, \\ \kappa_{\text{part}}, & a(y) = \text{partial agonist}, \\ 1, & a(y) = \text{antagonist}, \\ 0, & a(y) = \text{negative}, \end{cases}$$

where $\kappa_{\text{part}} \in (0, 1)$ is a hyperparameter reflecting lower confidence in partial agonism.

## 4.5 Direction Oracle

We introduce a Direction Oracle $f_\phi : \mathcal{A}^L \times \mathcal{X} \to [-1, 1]$, parameterized by $\phi$, that predicts the direction of transition bias. The predicted direction is defined as:

$$\hat{d}(y) = \text{sign}(f_\phi(y, x)). \tag{15}$$

Given a target protein sequence $x$ and a peptide binder $y$, the representations are obtained through pre-trained encoders:

$$\mathbf{h}_x = \text{Pool}(\mathcal{E}_x(x)), \quad \mathbf{h}_y = \text{Pool}(\mathcal{E}_y(y)), \tag{16}$$

where $\mathbf{h}_x \in \mathbb{R}^d$ and $\mathbf{h}_y \in \mathbb{R}^d$ denote the pooled embeddings for the target and binder. The fused representation is computed via a gated mechanism followed by an MLP:

$$\mathbf{z} = \mathbf{g} \odot \mathbf{h}_x + (1 - \mathbf{g}) \odot \mathbf{h}_y, \quad f_\phi(y, x) = \text{MLP}(\mathbf{z}), \tag{17}$$

where $\mathbf{g} = \sigma(\mathbf{W}_g[\mathbf{h}_x; \mathbf{h}_y] + \mathbf{b}_g)$ is a learned gating vector and $\odot$ denotes element-wise multiplication.

The oracle minimizes a weighted binary classification loss:

$$\mathcal{L}_{\text{dir}}(\phi) = \mathbb{E}_{(x,y,d)\sim\mathcal{D}} \left[ \kappa(y) \log(1 + \exp(-d \cdot f_\phi(y, x))) \right],$$

where $d \in \{-1, +1\}$ denotes the ground-truth direction and $\kappa(y)$ is a sample-dependent weight.

### 4.6 Contrastive Directional Representation

Let $h_\theta(y) \in \mathbb{R}^m$ denote a sequence representation extracted from the MDLM by mean-pooling the final-layer hidden states across all sequence positions. To enforce separation between directional classes in representation space, we define positive and negative index sets:

$$\mathcal{P} = \{(i,j) : d(y_i) = d(y_j), \ \kappa(y_i)\kappa(y_j) > 0\}, \quad (18)$$
$$\mathcal{N} = \{(i,j) : d(y_i) \neq d(y_j), \ \kappa(y_i)\kappa(y_j) > 0\}. \quad (19)$$

A margin-based contrastive loss is defined as:

$$\begin{aligned}
\mathcal{L}_{\text{ctr}}(\theta) = &\sum_{(i,j)\in\mathcal{P}} \|h_\theta(y_i) - h_\theta(y_j)\|_2^2 \\
&+ \sum_{(i,j)\in\mathcal{N}} \max(0, m - \|h_\theta(y_i) - h_\theta(y_j)\|_2)^2,
\end{aligned} \quad (20)$$

where $m > 0$ is a margin hyperparameter. Negative (non-binding) samples are excluded from this loss.

### 4.7 Incorporating Target Binding Affinity via Gating

Directional allosteric control is only meaningful for sequences that actually bind to the target protein. To ensure that directional supervision is applied to plausible binders, we incorporate a pre-trained peptide-protein affinity predictor as a soft gate within the reward function.

Let $g_\psi(y, x) \in [0, 1]$ denote a pre-trained affinity model that predicts the probability that peptide $y$ binds target protein $x$. Given a desired direction $d^\star \in \{+1, -1\}$, we define the gated reward as:

$$R(y; d^\star, x) = g_\psi(y, x) \cdot \sigma\left(\frac{d^\star \cdot f_\phi(y, x)}{\tau}\right), \quad (21)$$

where $\sigma$ denotes the sigmoid function and $\tau > 0$ is a temperature coefficient. Intuitively, the gated reward assigns high scores only to sequences that both bind the target protein (high $g_\psi$) and bias state transitions in the desired direction (high $d^\star \cdot f_\phi$, i.e., $\text{sign}(f_\phi)$ matches the requested $d^\star$). This formulation ensures that sequences predicted not to bind contribute negligible reward regardless of their directional score, while sequences predicted to bind are ranked according to their directional effect. Crucially, binding affinity acts as a *gate* to filter implausible candidates rather than a

quantity to be maximized: stronger binders do not necessarily induce desired state changes, so directional control must be decoupled from raw affinity rather than traded off against it on a Pareto frontier.

The resulting reward-tilted target distribution becomes:

$$p^\star(y \mid d^\star, x) \ \propto \ p_{\theta_0}(y) \exp\left(\frac{R(y; d^\star, x)}{\alpha}\right), \quad (22)$$

where $\alpha > 0$ is the regularization coefficient controlling the strength of deviation from the pre-trained prior.

Following the trajectory-level importance weighting framework from Section 3.3, the unnormalized log importance weight for a trajectory $y_{0:1}$ with final sequence $y = y_1$ is:

$$\begin{aligned}
\log \tilde{w}(y_{0:1}) = &\frac{R(y; d^\star, x)}{\alpha} \\
&+ \sum_{n=1}^{T} \sum_{\ell: y_{t_{n-1}}^\ell \neq y_{t_n}^\ell} \log \frac{p_{\theta_0}(y_{t_{n-1}}^\ell \mid y_{t_n}^{\text{UM}})}{p_{\bar{\theta}}(y_{t_{n-1}}^\ell \mid y_{t_n}^{\text{UM}})},
\end{aligned} \quad (23)$$

where $p_{\theta_0}$ and $p_{\bar{\theta}}$ denote the pre-trained model and proposal policy. At inference, a target protein $x$ and direction $d^\star \in \{+1, -1\}$ (agonist/antagonist) are provided. These inputs condition the reward function alone, keeping the generative backbone target-agnostic at the architectural level. We emphasize that this does not mean the backbone is unaffected by the target or direction: information about $x$ and $d^\star$ enters the policy through the reward signal during fine-tuning (Eq. 5), which reweights the WDCE loss and thus updates $\theta$, analogous to how preference signals in RLHF shape a language-model policy without being supplied as architectural inputs at inference time.

### 4.8 Amortized Fine-Tuning Objective

Direct sampling from $p^\star$ is intractable. We therefore learn a new parameterization $p_\theta(y)$ via amortized fine-tuning. Let $\mathcal{B}$ denote a replay buffer of sequences approximately sampled from $p^\star$ via tree search. The WDCE objective is:

$$\begin{aligned}
\mathcal{L}_{\text{WDCE}}(\theta) = &\mathbb{E}_{y\sim\mathcal{B}} \ \mathbb{E}_{t, y_t\sim q_t(\cdot|y)} \Big[ w(y) \\
&\sum_{\ell: y_t^\ell = \boldsymbol{M}} -\log p_\theta(y_\ell \mid y_t^{\text{UM}}, t) \Big],
\end{aligned} \quad (24)$$

where $\mathcal{B}$ is the replay buffer of candidate sequences populated by trajectory-aware tree search, $q_t(\cdot \mid y)$ is the forward masking process applied to sample a partially corrupted $y_t$ at time $t$, $y_t^{\text{UM}}$ is the unmasked context, and $w(y)$ is a per-sequence importance weight defined in Eq. 23. Intuitively, this objective performs standard masked-token denoising on buffer samples, but reweights each sample by $w(y)$ so that high-reward sequences (good binders that match the desired direction) contribute more gradient signal than low-reward ones, internalizing the gated reward into the model's

distribution. This softmax normalization over the buffer converts unnormalized log-weights into valid importance weights. Samples with $\kappa(y) = 0$ (non-binders) contribute no gradient.

To prevent collapse and preserve the pre-trained prior, we include a regularization term:

$$\mathcal{L}_{\text{reg}}(\theta) = \text{KL}(p_\theta \,\|\, p_{\theta_0}). \quad (25)$$

The full fine-tuning objective is:

$$\min_\theta \; \left\{ \mathcal{L}_{\text{WDCE}}(\theta) + \lambda_{\text{ctr}} \, \mathcal{L}_{\text{ctr}}(\theta) + \lambda_{\text{reg}} \, \mathcal{L}_{\text{reg}}(\theta) \right\}, \quad (26)$$

where $\lambda_{\text{ctr}}, \lambda_{\text{reg}} \geq 0$ are hyperparameters. At generation time, users specify a target protein $x$ and a desired direction $d^\star \in \{+1, -1\}$ corresponding to agonist or antagonist behavior.

### 4.9 Design Task

The directional allosteric design task is formally defined as:

**Design Task:** Given a desired direction $d^\star$, generate binder sequences whose induced transition asymmetry biases function toward the specified direction, without regressing kinetic rates or stabilizing endpoint states.

This formulation treats directionality of state transitions as the primary generative objective and defines a fully amortized procedure for incorporating coarse functional supervision into discrete sequence generation.

## 5 Results

We designed experiments to test whether modeling binder action as a sequence-conditioned transition operator captures forms of allosteric control that are inaccessible to equilibrium- and structure-centric design methods. Our evaluation addresses three questions: (1) whether directionally fine-tuned generators induce non-reversible transition behavior; (2) whether directional control can be achieved independently of binding affinity; and (3) whether the framework supports targeted control over specific transition directions rather than global perturbations.

### 5.1 Experimental Settings

**Dataset.** We curated data from the IUPHAR/BPS Guide to Pharmacology database (Harding et al., 2026), classifying entries as agonist or antagonist based on bioactivity. Interacting residues on ligands were extracted using PeptiDerive (Sedan et al., 2016) and converted to canonical SMILES for downstream processing. To balance agonist and antagonist samples, we generated synthetic antagonist

ligands via RFDiffusion (Watson et al., 2023) for targets with known agonists. After redundancy removal using MMseqs2 (Steinegger & Söding, 2017), the final dataset comprises 1,446 training and 336 held-out test samples.

The data split for TD3B training, validation, and test sets follows that of the Direction Oracle. Within the Direction Oracle training set, we further partitioned the data into training and validation subsets at an 8:1 ratio based on clustering. We filtered out binder sequences with residue counts outside the range of [16, 128], as well as targets containing only a single direction type (agonist or antagonist), to prevent direction bias toward specific targets. This preprocessing yields 130, 34, and 88 bidirectional target-binder pairs for the training, validation, and test sets, respectively.

**Evaluation Metrics.** For the Direction Oracle performance, we report Accuracy, Precision, Recall, and F1 Score to evaluate discriminative capability. To assess directional accuracy, inter-direction transitions, and designed binder affinity, we introduce direction-specific metrics for both agonist and antagonist modes: Affinity ($d^* = 1$), Affinity ($d^* = -1$), Direction Accuracy ($d^* = 1$), and Direction Accuracy ($d^* = -1$). Additionally, we report the gated reward to evaluate overall model training effectiveness

**Setup.** We adopt the pre-trained MDLM weights from PepTune (Tang et al., 2025a) as our base model. During finetuning, we employ tree search for buffer generation following TR2-D2 (Tang et al., 2025c), regenerating binder data for multiple targets every $k$ iterations with a first-in-first-out buffer to mitigate catastrophic forgetting. Binding affinity is estimated via a pre-trained predictor (Tang et al., 2025a; Zhang et al., 2026), trained on the PepLand dataset (Zhang et al., 2025) to produce a continuous, normalized affinity score (combining $K_d$, $K_i$, and $IC_{50}$), where higher values indicate stronger binding and a one-unit increase corresponds to an approximate tenfold change in binding strength. We note that RFDiffusion is used in this paper only as an external baseline (Sec. 5.3) and for synthetic-antagonist data augmentation; it is *not* part of the TD3B inference pipeline, and TD3B applies no docking or structural filtering to its generated sequences. Additional details on ground-truth affinity sources and the structural inputs used for the RFDiffusion baseline are provided in Appendix B.3. For the Direction Oracle, protein target sequences are encoded using ESM2 (Lin et al., 2023), and binder sequences are tokenized using the SPE tokenizer (Tang et al., 2025c). During generation, we apply Algorithm 2 for weighted sampling, generating 8 samples per direction for each target.

### 5.2 Direction Oracle for Protein-Peptide Binding

To ensure sufficient exploration space for both tree search and model training, we first demonstrate that the Direction

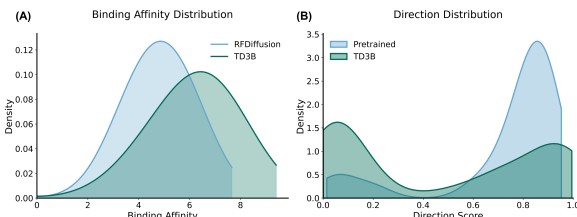

*Figure 3.* Direction and affinity distribution of TD3B. **(A)** Binding affinity comparison vs. RFDiffusion. **(B)** Direction comparison vs. pre-trained PepTune.

Oracle can accurately identify the directionality of binders across diverse targets and varying sequence lengths, thereby providing reliable guidance during training. As shown in Table 1, the Direction Oracle achieves strong classification performance across all metrics.

*Table 1.* Binary classification performance of the Direction Oracle.

|  | Accuracy | Precision | Recall | F1 |
|---|---|---|---|---|
| Dir. Oracle | 0.93 | 0.90 | 0.91 | 0.90 |

### 5.3 Assessing Directional Asymmetry of Learned Transition Operators

Binding affinity is a prerequisite for characterizing agonists and antagonists (bru, 1980). To validate TD3B, we compared the predicted binding affinity of TD3B-generated samples against those designed by structure-based RFDiffusion (Watson et al., 2023). Figure 3A shows TD3B achieves higher predicted normalized affinity, demonstrating the effectiveness of gated reward fine-tuning.

We evaluated the directional behavior of peptides generated under agonist- versus antagonist-directed finetuning by measuring samples from the pre-trained (unconditioned) generator and TD3B using the Direction Oracle. As shown in Figure 3B, the pre-trained model generates predominantly agonist-biased binders with low confidence and lacks directional control. In contrast, TD3B produces distributions with higher confidence across both directions and enables explicit control over transition directionality, though we note asymmetry remains (see Table 2).

### 5.4 Comparison to Static and Predictive Baselines

We test whether TD3B can break the directional symmetry between activation and inhibition. Since pre-trained discrete diffusion models lack direction conditioning, we compare training-free guided diffusion against tree-search-based fine-tuning. Table 2 shows TD3B achieves the highest gated reward, confirming its effectiveness for direction-specific generation. Direction accuracy is asymmetric across agonist and antagonist modes, reflecting the stronger antagonist

signal in training data and the pre-trained model's inherent agonist bias. Finetuning-based methods achieve higher affinity and better directional balance than guidance-only approaches. Compared to TR2-D2, TD3B adds weighted sampling to exploit high-potential samples and a contrastive loss to separate distributions in latent space, enabling directional understanding beyond iterative optimization. Ablations on loss components and weighted sampling are in Appendix C.3.

As shown in Table 2, both $\mathcal{L}_{\mathrm{ctr}}$ and $\mathcal{L}_{\mathrm{reg}}$ are necessary. Without $\mathcal{L}_{\mathrm{ctr}}$, agonist and antagonist accuracies collapse to nearly identical values, indicating that the latent space no longer separates the two directions. Without $\mathcal{L}_{\mathrm{reg}}$, the model drifts toward the agonist-biased pre-trained prior (Figure 3B): agonist accuracy rises while antagonist accuracy becomes unstable across seeds. The KL term thus preserves the distributional capacity needed to reach the antagonist mode, which lies further from the prior. Per-target analysis (Appendix C.2) further shows that remaining agonist failures cluster on a small number of targets with limited training signal, suggesting asymmetric loss weighting as a natural future direction.

*Table 2.* Comparison of model performance across various metrics. Best results are in bold. CG: Classifier Guidance; SMC: Sequential Monte Carlo; TDS: Twisted Diffusion Sampler (Wu et al., 2023).

| Method | Aff($d^\star = +1$) | Aff($d^\star = -1$) | DA($d^\star = +1$) | DA($d^\star = -1$) | Gated |
|---|---|---|---|---|---|
| Pre-trained | $4.80_{\pm 0.01}$ | $4.82_{\pm 0.01}$ | $0.822_{\pm 0.010}$ | $0.174_{\pm 0.010}$ | $2.06_{\pm 0.32}$ |
| CG | $4.80_{\pm 0.00}$ | $4.81_{\pm 0.01}$ | $0.816_{\pm 0.005}$ | $0.176_{\pm 0.014}$ | $2.38_{\pm 0.02}$ |
| SMC | $4.83_{\pm 0.01}$ | $4.78_{\pm 0.01}$ | $0.879_{\pm 0.011}$ | $0.470_{\pm 0.011}$ | $3.18_{\pm 0.31}$ |
| TDS | $4.82_{\pm 0.02}$ | $4.79_{\pm 0.02}$ | $0.841_{\pm 0.032}$ | $0.235_{\pm 0.028}$ | $2.42_{\pm 0.07}$ |
| PepTune | $5.02_{\pm 0.09}$ | $5.00_{\pm 0.08}$ | $0.618_{\pm 0.230}$ | $0.437_{\pm 0.142}$ | $2.61_{\pm 0.28}$ |
| TR2-D2 | $5.89_{\pm 0.15}$ | $5.89_{\pm 0.19}$ | $0.178_{\pm 0.127}$ | $0.875_{\pm 0.053}$ | $3.36_{\pm 0.24}$ |
| TD3B w/o $\mathcal{L}_{\mathrm{ctr}}$ | $5.33_{\pm 0.48}$ | $5.35_{\pm 0.61}$ | $0.788_{\pm 0.190}$ | $0.788_{\pm 0.196}$ | $4.01_{\pm 0.10}$ |
| TD3B w/o $\mathcal{L}_{\mathrm{reg}}$ | $5.28_{\pm 0.47}$ | $5.16_{\pm 0.26}$ | $\mathbf{0.938_{\pm 0.010}}$ | $0.471_{\pm 0.168}$ | $3.84_{\pm 0.14}$ |
| **TD3B (Ours)** | $\mathbf{6.00_{\pm 0.02}}$ | $\mathbf{6.32_{\pm 0.02}}$ | $0.795_{\pm 0.017}$ | $\mathbf{1.000_{\pm 0.000}}$ | $\mathbf{5.33_{\pm 0.03}}$ |

### 5.5 Targeted Control of Transition Asymmetry

We next assessed TD3B's targeted control over specific transitions without global dynamic perturbation. Conditioning on $d^\star \in \{+1, -1\}$, we verified that generated binders selectively bias transitions while maintaining high affinity. Table 3 shows *de novo* TD3B binders outperform length-matched wild-type references in affinity across all directions. Success rate, defined as the fraction of binders achieving both (i) superior predicted affinity to wild-type and (ii) correct Direction Oracle classification, reaches 61% and 100% for forward and backward transitions, respectively. This confirms directionality as a controllable objective rather than an incidental outcome of binding optimization.

### 5.6 Case Studies

We fine-tuned TD3B on two key GPCR targets (GLP1R and TAAR1) and selected top-ranked candidates for each. Complex structures were predicted using AlphaFold3, fol-

*Table 3.* Targeted control: Evaluation of functional specificity across transition objectives.

| Design Objective ($d^*$) | Affinity (WT) | Affinity (Transition) | Success Rate |
|---|---|---|---|
| Forward Transition ($d^* = 1$) | 4.66 | 5.81 | 0.61 |
| Reverse Transition ($d^* = -1$) | 4.99 | 6.06 | 1.00 |

lowed by binding-site detection and scoring with an external classifier. We first focused on GLP-1R, a clinically important metabolic receptor revolutionary for obesity, weight loss, and type 2 diabetes treatment where agonist activity is the primary therapeutic mechanism (Moiz et al., 2025). As reference, Figure 4A shows the AF3-predicted structure of GLP-1R with all 37 interacting residues at the binding interface of the endogenous GLP-1 hormone. TD3B-designed agonists engage key activation residues including Tyr148, Tyr152, Arg190, Lys197, Tyr205, Gln234, Trp297, Thr298, Arg299, and Asn300 (Liao & Tzen, 2022), with Arg299 and Asn300 being essential for full agonist activity (Lei et al., 2018) (Figure 4B). In contrast, TD3B-designed antagonists (Figure 4C) lack interactions with Arg299 and Asn300, consistent with their inability to activate the receptor. These results demonstrate that TD3B can design agonist and antagonist binders that selectively engage or avoid critical activation residues on the same receptor.

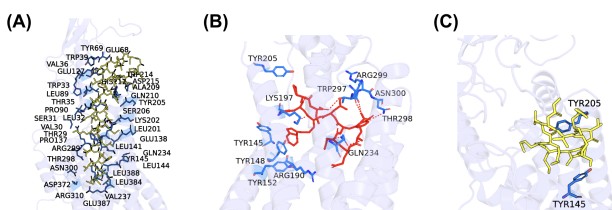

*Figure 4.* Evaluation of TD3B on GLP-1R. **(A)** Existing GLP-1 peptide hormone bound to GLP-1R. **(B)** TD3B-designed agonist bound GLP-1R. **(C)** TD3B-designed antagonist bound GLP-1R.

We next applied TD3B to the Orexin 1 Receptor (OX1R), a regulatory GPCR where antagonists treat insomnia and agonists show promise for narcolepsy (Scammell & Winrow, 2011; Nishino et al., 2000). Figure 5A shows the crystal structure of OX1R bound to the clinical antagonist suvorexant (PDB: 4ZJ8), revealing 14 key orthosteric residues. TD3B-designed agonists engage 5 conserved residues, while antagonists engage 6 residues (Figures 5B and C). Both ligands contact GLN126, a molecular switch that controls activation through hydrogen bonding with TYR348 (Karhu et al., 2019). The TD3B-designed antagonist engages 6 conserved residues, including GLN126, a contact pattern consistent with inactive-state stabilization by suvorexant (PDB: 4ZJ8) (Yin et al., 2016). The TD3B-designed agonist engages 5 residues with a distinct pattern that does *not* reinforce this inactive-state geometry, potentially per-

mitting the conformational flexibility required for activation. We acknowledge that this interpretation is based on static AlphaFold3-predicted complex structures; definitive mechanistic conclusions about activation will require molecular dynamics simulations and experimental mutagenesis, which we identify as a priority for future work. The substantial conservation with suvorexant's validated binding site nonetheless confirms orthosteric targeting and demonstrates TD3B's ability to design functionally distinct ligands for the same receptor pocket. Together, these results indicate that TD3B can generate functionally divergent binders for the same target by modulating transition directionality rather than binding affinity alone. More cases are shown in Appendix C.1.

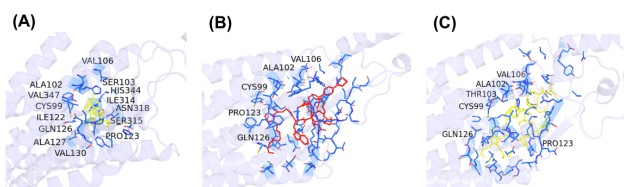

*Figure 5.* Evaluation of TD3B on OX1R. **(A)** Suvorexant bound to OX1R (PDB: 4ZJ8). **(B)** TD3B-designed agonist. **(C)** TD3B-designed antagonist.

# 6 Discussion

We introduce **TD3B**, a generative framework that formulates allosteric binder design through control over sequence-conditioned transition operators instead of optimization toward static states or equilibrium energies. By making directional asymmetry and non-reversibility explicit modeling targets, TD3B captures functional effects that structure-centric design algorithms and predictive dynamics models fail to address. Specifically, the transition-operator formalism in Sections 4.1-4.3 supplies a theoretical foundation without adding a parameterized component. TD3B never regresses the operator $Q^{(y)}$ and uses supervision that captures only the binary sign of $\Delta(y)$, which sidesteps the need for high-resolution kinetic measurements that remain difficult to obtain at scale. This perspective generalizes across modalities, supporting generative design wherever function arises from non-equilibrium state shifts. Our natural next steps include richer continuous-rate supervision from Markov state models, multi-state generalizations that capture biased agonism, and wet-lab validation on representative GPCR targets such as GLP-1R and OX1R through functional assays like cAMP and $\beta$-arrest recruitment. Together, these directions point toward generative models that learn to shape, instead of merely sample from, the functional dynamics of biomolecular systems.

## Acknowledgments

This research was supported by a grant from the High-throughput Institute for Discovery (HIT-ID) at the University of Pennsylvania to the lab of Pranam Chatterjee. The work described in this paper was also partially supported by the Research Grants Council of the Hong Kong Special Administrative Region, China, under Project T45-401/22-N.

## Impact Statement

This work develops a computational framework for generating protein-binding peptides with controlled agonist or antagonist behavior, which may support therapeutic discovery efforts where functional directionality is critical, such as receptor activation or inhibition. By separating binding from functional effect, the approach aims to reduce late-stage failures associated with ligands that bind but produce unintended signaling outcomes. At the same time, generative models for bioactive molecules carry risks, including off-target interactions or unanticipated biological activity. The methods presented here are intended as hypothesis-generation tools and require careful experimental validation, target-specific safety assessment, and responsible use before any therapeutic application.

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

# Appendix

# A   Theoretical Proofs

This appendix records basic guarantees for TD3B. We (i) justify exponential tilting as the unique solution of a KL-regularized improvement objective, (ii) characterize the population-optimal Direction Oracle under weighted logistic risk, (iii) relate weighted denoising cross-entropy to fitting a target (tilted) distribution, (iv) bound the effect of oracle approximation error on the induced tilted distribution, and (v) state a simple separability consequence of zero contrastive loss.

## A.1   Notation and Setup

We adopt notation from the main text.

- $\mathcal{Y} := \mathcal{A}^L$ denotes the finite sequence space.

- $p_0(y)$ denotes a fixed base distribution on $\mathcal{Y}$ (e.g., the pre-trained MDLM distribution $p_{\theta_0}$).

- $S : \mathcal{Y} \to \mathbb{R}$ denotes a score (objective) and $\alpha > 0$ a temperature.

- The (reward-)tilted distribution is

$$p^\star(y) := \frac{p_0(y)\exp(S(y)/\alpha)}{Z}, \qquad Z := \sum_{y' \in \mathcal{Y}} p_0(y')\exp(S(y')/\alpha). \tag{27}$$

- For the direction task, we use labels $d(y) \in \{+1, -1\}$, confidence weights $\kappa(y) \in [0, 1]$, and an oracle $f_\phi : \mathcal{Y} \to \mathbb{R}$. The direction-only score used for design is $S(y; d^\star) = d^\star f_\phi(y)$ for $d^\star \in \{+1, -1\}$.

- For distributions $P, Q$ on $\mathcal{Y}$ we write $\|P - Q\|_{\mathrm{TV}}$ for total variation and $\mathrm{KL}(P\|Q)$ for Kullback–Leibler divergence.

## A.2   Exponential Tilting as KL-Regularized Improvement

**Theorem A.1** (Exponential tilting solves KL-regularized improvement). *Fix a base distribution $p_0$ on $\mathcal{Y}$ and a score $S : \mathcal{Y} \to \mathbb{R}$. Consider the optimization problem over distributions $q$ on $\mathcal{Y}$:*

$$\max_{q \in \Delta(\mathcal{Y})} \left\{ \mathbb{E}_{Y \sim q}[S(Y)] - \alpha\,\mathrm{KL}(q\|p_0) \right\}, \qquad \alpha > 0. \tag{28}$$

*Then the unique maximizer is the tilted distribution $p^\star$ in (27).*

*Proof.* Since $\mathcal{Y}$ is finite, (28) is a strictly concave optimization problem in $q$. Introduce a Lagrange multiplier $\lambda$ for the constraint $\sum_y q(y) = 1$. The Lagrangian is

$$\mathcal{J}(q, \lambda) = \sum_y q(y)S(y) - \alpha \sum_y q(y) \log \frac{q(y)}{p_0(y)} + \lambda\left(\sum_y q(y) - 1\right). \tag{29}$$

Taking derivatives with respect to $q(y)$ and setting to zero gives

$$S(y) - \alpha\left(\log q(y) - \log p_0(y) + 1\right) + \lambda = 0, \tag{30}$$

so $\log q(y) = \log p_0(y) + S(y)/\alpha + c$ for a constant $c$. Hence

$$q(y) \propto p_0(y)\exp(S(y)/\alpha), \tag{31}$$

and normalization yields (27). Strict concavity implies uniqueness. $\square$

**Corollary A.1** (Limiting cases). *Let $p^\star$ be defined by (27). Then:*

1. *As $\alpha \to \infty$, $p^\star \to p_0$ in total variation.*

2. *As $\alpha \to 0^+$, $p^\star$ concentrates on $\arg\max_{y\in\mathcal{Y}} S(y)$ (within the support of $p_0$).*

*Proof.* Write $p^\star(y) = p_0(y)\exp(S(y)/\alpha)/Z$. As $\alpha \to \infty$, $\exp(S(y)/\alpha) \to 1$ uniformly, so $Z \to 1$ and $p^\star \to p_0$. As $\alpha \to 0^+$, the normalization is dominated by maximizers of $S$ because $\exp(S(y)/\alpha)$ is a softmax with temperature $\alpha$. $\quad\square$

---

**Proposition A.1** (Relative odds under tilting). *For any $y_1, y_2 \in \mathcal{Y}$ with $p_0(y_1), p_0(y_2) > 0$,*

$$\frac{p^\star(y_1)}{p^\star(y_2)} = \frac{p_0(y_1)}{p_0(y_2)} \exp\left(\frac{S(y_1) - S(y_2)}{\alpha}\right). \tag{32}$$

*In particular, if $p_0(y_1) = p_0(y_2)$ then the odds ratio depends only on $S(y_1) - S(y_2)$.*

---

*Proof.* Immediate from the definition (27) since the normalizer $Z$ cancels. $\quad\square$

**Binary direction specialization.** For $S(y; d^\star) = d^\star f(y)$ and $d^\star \in \{+1, -1\}$,

$$\frac{p^\star(y \mid +1)}{p^\star(y \mid -1)} = \frac{Z_-}{Z_+} \exp\left(\frac{2f(y)}{\alpha}\right), \qquad Z_\pm := \sum_y p_0(y)\exp(\pm f(y)/\alpha). \tag{33}$$

Thus larger oracle score $f(y)$ implies larger posterior odds of the $+1$-tilt relative to the $-1$-tilt.

## A.3 Population Optimality of the Direction Oracle

We formalize the direction oracle as a weighted logistic risk minimizer.

---

**Theorem A.2** (Bayes-optimal oracle under weighted logistic loss). *Let $(Y, D)$ be a random pair where $Y \in \mathcal{Y}$ and $D \in \{+1, -1\}$. Let $\kappa : \mathcal{Y} \times \{+1, -1\} \to [0, \infty)$ be a measurable weight (in TD3B, $\kappa$ encodes down-weighting of partial agonists and exclusion of non-binders). Consider minimizing*

$$\mathcal{R}(f) := \mathbb{E}[\kappa(Y, D) \log(1 + \exp(-Df(Y)))] \tag{34}$$

*over all functions $f : \mathcal{Y} \to \mathbb{R}$. Define*

$$\eta_+(y) := \mathbb{E}[\kappa(Y, D)\mathbf{1}\{D = +1\} \mid Y = y], \qquad \eta_-(y) := \mathbb{E}[\kappa(Y, D)\mathbf{1}\{D = -1\} \mid Y = y]. \tag{35}$$

*If $\eta_+(y) + \eta_-(y) > 0$, the pointwise minimizer satisfies*

$$f^\star(y) = \log \frac{\eta_+(y)}{\eta_-(y)}. \tag{36}$$

---

*Proof.* Fix $y \in \mathcal{Y}$ and write the conditional risk (up to an additive constant independent of $f(y)$) as

$$r_y(u) = \eta_+(y) \log(1 + \exp(-u)) + \eta_-(y)\log(1 + \exp(u)), \qquad u := f(y). \tag{37}$$

This is strictly convex in $u$. Differentiate and set to zero:

$$r'_y(u) = -\eta_+(y)\frac{1}{1 + \exp(u)} + \eta_-(y)\frac{\exp(u)}{1 + \exp(u)} = 0. \tag{38}$$

Multiplying by $1 + \exp(u)$ gives $-\eta_+(y) + \eta_-(y)\exp(u) = 0$, hence $\exp(u) = \eta_+(y)/\eta_-(y)$ and (36) follows. $\quad\square$

**Remark.** Theorem A.2 shows that, in the population limit, a weighted logistic oracle estimates a (weighted) log-odds function. This makes exponential tilting with $S(y; d^\star) = d^\star f(y)$ a principled way to bias generation toward one directional class.

## A.4 Weighted Denoising Cross-Entropy Fits a Target Distribution

We formalize the effect of WDCE as fitting an MDLM to a reweighted data distribution.

**Lemma A.1** (Weighted risk equals unweighted risk under a reweighted distribution). *Let $r$ be any distribution on $\mathcal{Y}$ and let $w : \mathcal{Y} \to [0, \infty)$ be a weight function with $0 < \mathbb{E}_{Y \sim r}[w(Y)] < \infty$. Define the normalized reweighted distribution*

$$\pi(y) := \frac{r(y)w(y)}{\mathbb{E}_{Y \sim r}[w(Y)]}. \tag{39}$$

*Then for any nonnegative loss $\ell(y)$,*

$$\mathbb{E}_{Y \sim r}[w(Y)\ell(Y)] = \mathbb{E}_{Y \sim r}[w(Y)] \cdot \mathbb{E}_{Y \sim \pi}[\ell(Y)]. \tag{40}$$

*Proof.* By definition, $\mathbb{E}_{Y \sim \pi}[\ell(Y)] = \sum_y \pi(y)\ell(y) = \frac{1}{\mathbb{E}_r[w]} \sum_y r(y)w(y)\ell(y)$. Rearrange. $\square$

**Theorem A.3** (Population optimality of WDCE denoisers). *Fix a corruption kernel family $q_t(y_t \mid y)$ and a time sampling distribution over $t \in [0, 1]$. Let $\pi$ be a target distribution on $\mathcal{Y}$ and define the joint $(y, y_t)$ by $y \sim \pi$ and $y_t \sim q_t(\cdot \mid y)$. Consider the MDLM denoising objective*

$$\mathcal{L}_\pi(\theta) = \mathbb{E}_{t,y,y_t}\left[ \sum_{\ell:y_t^\ell=M} -\log p_\theta\left(y^\ell \mid y_t^{\mathrm{UM}}, t\right) \right]. \tag{41}$$

*Then for each time $t$ and each masked position $\ell$, the minimizer satisfies*

$$p_{\theta^\star}\left(y^\ell = a \mid y_t^{\mathrm{UM}}, t\right) = \mathbb{P}_\pi\left(y^\ell = a \mid y_t^{\mathrm{UM}}, t\right) \qquad \forall a \in \mathcal{A}, \tag{42}$$

*that is, the optimal denoiser recovers the true conditional under $\pi$.*

*Proof.* Fix $t$ and condition on the context $C := (y_t^{\mathrm{UM}}, t)$ and the event that position $\ell$ is masked. The inner term in (41) is the cross-entropy between the true conditional distribution of $y^\ell$ given $C$ and the model distribution $p_\theta(\cdot \mid C)$. Cross-entropy is minimized uniquely by matching the true conditional. Taking expectation over contexts yields the result. $\square$

**Connection to reward tilting.** If the proposal distribution $r$ is $p_0$ and weights are

$$w(y) = \exp\left(\frac{S(y_1)}{\alpha}\right) \cdot \prod_{n=1}^T \frac{p_0(y_{t_{n-1}} \mid y_{t_n})}{p_{\bar{\theta}}(y_{t_{n-1}} \mid y_{t_n})}, \tag{43}$$

then the reweighted distribution $\pi$ in Lemma A.1 equals the tilted distribution $p^\star$ in (27). Thus WDCE is (in the population limit) standard MDLM training under the tilted target distribution.

## A.5 Stability of Tilting Under Oracle Approximation Error

**Theorem A.4** (Tilt robustness under bounded score error). *Let $S^\star : \mathcal{Y} \to \mathbb{R}$ be an ideal score and let $S : \mathcal{Y} \to \mathbb{R}$ satisfy*

$$\sup_{y \in \mathcal{Y}} |S(y) - S^\star(y)| \leq \varepsilon. \tag{44}$$

*Let $p^\star$ and $\widetilde{p}$ be the corresponding tilted distributions built from $(p_0, S^\star)$ and $(p_0, S)$ with the same temperature $\alpha > 0$. Then*

$$\mathrm{KL}(\widetilde{p}\|p^\star) \leq \frac{2\varepsilon}{\alpha}, \qquad \mathrm{KL}(p^\star\|\widetilde{p}) \leq \frac{2\varepsilon}{\alpha}, \tag{45}$$

*and therefore, by Pinsker's inequality,*

$$\|\widetilde{p} - p^\star\|_{\mathrm{TV}} \leq \sqrt{\frac{\varepsilon}{\alpha}}. \tag{46}$$

*Proof.* Write $S = S^\star + \delta$ with $|\delta(y)| \leq \varepsilon$. Then

$$\widetilde{p}(y) = \frac{p_0(y)\exp((S^\star(y) + \delta(y))/\alpha)}{\widetilde{Z}} = p^\star(y) \frac{\exp(\delta(y)/\alpha)}{\mathbb{E}_{Y \sim p^\star}[\exp(\delta(Y)/\alpha)]}. \tag{47}$$

Since $\exp(\delta/\alpha) \in [\exp(-\varepsilon/\alpha), \exp(\varepsilon/\alpha)]$, the normalizer ratio satisfies

$$\mathbb{E}_{p^\star}[\exp(\delta/\alpha)] \in [\exp(-\varepsilon/\alpha), \exp(\varepsilon/\alpha)]. \tag{48}$$

Hence for all $y$,

$$\log \frac{\widetilde{p}(y)}{p^\star(y)} = \frac{\delta(y)}{\alpha} - \log \mathbb{E}_{p^\star}[\exp(\delta/\alpha)] \in \left[-\frac{2\varepsilon}{\alpha}, \frac{2\varepsilon}{\alpha}\right]. \tag{49}$$

Taking expectation under $\widetilde{p}$ yields $\mathrm{KL}(\widetilde{p}\|p^\star) \leq 2\varepsilon/\alpha$. The reverse KL bound follows symmetrically by swapping roles of $(S, S^\star)$. Pinsker's inequality gives the TV bound. $\square$

## A.6 A Separability Consequence of Zero Contrastive Loss

**Proposition A.2** (Zero margin-contrastive loss implies linear separability). *Let $\{(y_i, d_i)\}_{i=1}^N$ be labeled samples with $d_i \in \{+1, -1\}$ and embeddings $h(y_i) \in \mathbb{R}^m$. Consider the margin-contrastive loss*

$$\mathcal{L}_{\mathrm{ctr}} = \sum_{(i,j):d_i=d_j} \|h(y_i) - h(y_j)\|_2^2 + \sum_{(i,j):d_i \neq d_j} \max(0, m_0 - \|h(y_i) - h(y_j)\|_2)^2. \tag{50}$$

*If $\mathcal{L}_{\mathrm{ctr}} = 0$, then there exist $u_+, u_- \in \mathbb{R}^m$ such that $h(y_i) = u_{d_i}$ for all $i$ and $\|u_+ - u_-\|_2 \geq m_0$. In particular, the classes are linearly separable by a hyperplane with margin at least $m_0/2$.*

*Proof.* If $\mathcal{L}_{\mathrm{ctr}} = 0$, then for any pair $(i, j)$ with $d_i = d_j$ we must have $\|h(y_i) - h(y_j)\|_2^2 = 0$, hence all embeddings within a class are identical. Denote the two class prototypes by $u_+$ and $u_-$. For any pair with $d_i \neq d_j$, the hinge term being zero implies $\|u_+ - u_-\|_2 \geq m_0$.

Define $w := u_+ - u_-$ and $b := -\frac{1}{2}\langle w, u_+ + u_-\rangle$. Then

$$\langle w, u_+\rangle + b = \frac{1}{2}\|u_+ - u_-\|_2^2 \geq \frac{1}{2}m_0^2, \qquad \langle w, u_-\rangle + b = -\frac{1}{2}\|u_+ - u_-\|_2^2 \leq -\frac{1}{2}m_0^2, \tag{51}$$

so the hyperplane $\{z : \langle w, z\rangle + b = 0\}$ separates the two prototypes. The (geometric) margin is at least $\|u_+ - u_-\|_2/2 \geq m_0/2$. $\square$

# B  Implementation and Dataset Details

## B.1  Direction Oracle Training

The Direction Oracle is trained for 20 epochs using AdamW optimization with a learning rate of $10^{-5}$ and batch size of 16, minimizing cross-entropy loss. pre-trained encoders remain frozen throughout training; only projection layers, self-attention and cross-attention modules, and the two-layer MLP classifier head are optimized. Due to the limited size of available labeled data, we train on the full training split without validation and evaluate performance exclusively on an independent held-out test set.

## B.2  TD3B Finetuning

For tree search-based sampling, we employ trajectory-aware tree search with 20 iterations and 24 children per node, sampling 4 targets per iteration with a buffer size of 32 candidates per target. A replay buffer of 2000 samples with FIFO replacement is maintained to mitigate catastrophic forgetting. Training uses a batch size of 4 with gradient accumulation over 4 steps, a learning rate of $5 \times 10^{-5}$, and 4 WDCE replicates per sample. The KL regularization coefficient $\lambda_{\mathrm{reg}}$ is set to 0.5, and tree search resampling is performed every 10 epochs. Training is conducted on 8 NVIDIA A100 GPUs using PyTorch DDP with synchronized buffer aggregation.

## B.3  Affinity Sources and Structural Inputs for Baselines

**Ground-truth binding affinities.**  Ground-truth binding affinities used for evaluation in Section 5.5 are taken from the SKEMPI 2.0 database (Jankauskaitė et al., 2019), which compiles experimentally measured dissociation constants ($K_d$) from surface plasmon resonance and spectroscopic techniques and converts them to binding free energy changes ($\Delta\Delta G$). These measurements provide an independent, experimentally grounded reference, as opposed to the pre-trained PepLand-based affinity predictor used internally for fine-tuning rewards.

**Structural inputs to the RFDiffusion baseline.**  For the RFDiffusion baseline reported in Section 5.3 and Figure 3A, all structural inputs are experimentally determined PDB-deposited structures with validated agonist or antagonist activity for the corresponding target. Using experimentally resolved rather than model-predicted structures avoids confounding the comparison with potential conformational biases from upstream prediction (e.g., a structure predictor's preference for an agonist- or antagonist-bound state). RFDiffusion was used here strictly as an external baseline and, separately, for synthetic-antagonist data augmentation during dataset construction; it is *not* part of the TD3B pipeline at training or inference time, and TD3B applies no docking, structural filtering, or energy minimization to its outputs.

## B.4  Data Leakage Analysis

A natural concern with using an externally pre-trained affinity predictor (trained on the PepLand dataset (Zhang et al., 2025)) is whether any pair-level overlap exists with our IUPHAR/BPS-derived training and test sets, which could artificially inflate apparent generalization. We performed three checks comparing our 130 training and 88 test bidirectional pairs against the 2,110 entries in PepLand: (i) exact peptide sequence match, (ii) exact UniProt-ID match for the target, and (iii) $(target, peptide)$ pair match. We additionally screened for approximate matches at $\geq 80\%$ pairwise sequence identity. Results are summarized in Table 4.

*Table 4.* Pair-level data leakage analysis between our IUPHAR/BPS-derived dataset and PepLand. No exact peptide–target pair appears in both datasets.

| Check | Test set | Train set |
|---|---|---|
| Exact peptide match | 0 | 0 |
| Exact target match | 0 | 0 |
| Exact pair / UniProt-ID pair | 0 | 0 |
| Approximate peptide ($\geq 80\%$ id.) | 1 (diff. target) | 3 (substring) |

The single approximate peptide match (P50984; $87.5\%$ identity, two-residue difference) is paired with entirely different targets in the two datasets (ours: P30532/P32297; PepLand: Q8WSF8). Since the affinity predictor is conditioned on both target and binder, no pair-level binding information transfers, and we conclude that the apparent generalization of TD3B is

not attributable to leakage through the affinity predictor.

## C  Extended Experimental Results

### C.1  Case Studies on Additional Protein Targets

We further applied TD3B to TAAR1, a neuromodulatory GPCR implicated in dopaminergic and serotonergic signaling and strongly linked to schizophrenia, where both agonists and antagonists are of pharmacological interest (Dedic et al., 2021). As shown in Figure C1, TD3B-generated agonists and antagonists for TAAR1 exhibit distinct binding modes. Similar to the activatory contacts of T1AM (3-iodothyronamine), a small molecule TAAR1 agonist shown in PDB 8JLN (Figure C1A) (Xu et al., 2023), the designed agonist preferentially engages with a compact orthosteric pocket spanning TM3, TM5, TM6, and TM7, (Jones et al., 2020) including residues associated with receptor activation (Huang et al., 2025) (Figure C1B), and the designed antagonist occupies a broader binding region that partially overlaps the orthosteric site but selectively avoids key activation-associated transmembrane contacts (Figure C1C).

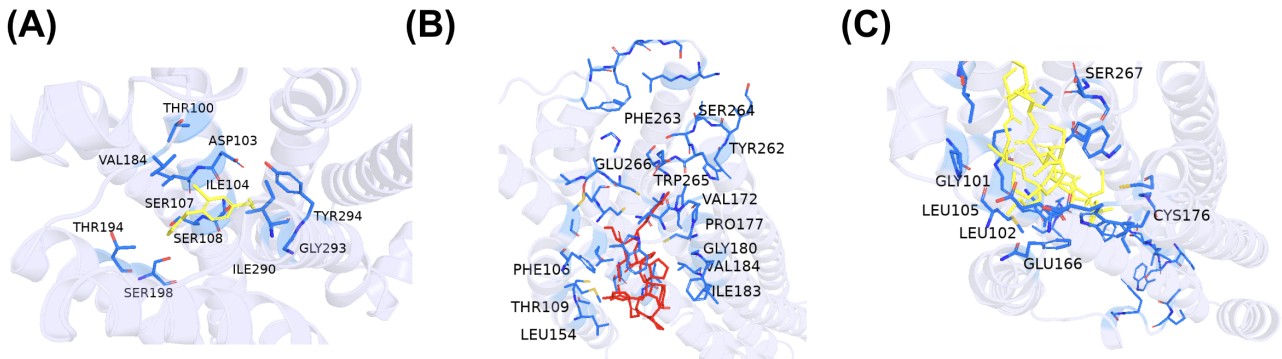

*Figure C1.* Evaluation of TD3B on TAAR1. **(A)** Existing TAAR1 agonist bound to TAAR1. **(B)** TD3B-designed agonist bound TAAR1. **(C)** TD3B-designed antagonist bound TAAR1.

### C.2  Per-Target Performance Breakdown

Table 5 reports per-target gated reward and direction accuracies for TD3B with 3-seed evaluation. Cross-seed variance is small ($\pm 0.03$ on aggregated gated reward), confirming that training is stable; the residual per-target variance is concentrated on a small number of targets where agonist generation fails ($\mathrm{DA}(d^\star = +1) = 0$ on 3/10 targets while $\mathrm{DA}(d^\star = -1) = 1.00$ remains intact). This pattern is consistent with the agonist/antagonist asymmetry discussed in App. C.3: targets with limited agonist training signal are disproportionately affected, while antagonist generation is essentially saturated. Notably, on the unseen target P35368, TD3B reaches $\mathrm{DA} = 1.00$ in both directions, indicating that the asymmetry is target-specific rather than fundamental.

*Table 5.* Per-target gated reward and direction accuracies for TD3B (3-seed evaluation).

| Target (UniProt) | $n$ | $\mathrm{Aff}(d^\star = +1)$ | $\mathrm{Aff}(d^\star = -1)$ | $\mathrm{DA}(d^\star = +1)$ | $\mathrm{DA}(d^\star = -1)$ | Gated |
|---|---|---|---|---|---|---|
| P30532 | 200 | 7.05 | 8.04 | 0.00 | 1.00 | 4.00 |
| P32239 | 200 | 6.89 | 7.22 | 1.00 | 1.00 | 7.00 |
| P32246 | 200 | 6.36 | 7.15 | 1.00 | 1.00 | 6.53 |
| P32297 | 200 | 6.24 | 7.29 | 0.00 | 1.00 | 3.72 |
| P35368 | 200 | 6.19 | 6.94 | 1.00 | 1.00 | 6.08 |
| P41587 | 200 | 6.97 | 6.91 | 0.00 | 0.00 | 0.97 |
| P51677 | 200 | 7.94 | 6.97 | 1.00 | 1.00 | 7.35 |
| P54282 | 200 | 4.52 | 4.88 | 0.00 | 1.00 | 2.43 |
| Q63447 | 100 | 5.20 | – | 1.00 | – | 5.16 |
| Q6W5P4 | 200 | 5.03 | 6.67 | 1.00 | 1.00 | 5.32 |

## C.3 Ablation Studies

**Weighted sampling.** TD3B differs from TR2-D2 along two axes: (i) training-time innovations (gated reward + contrastive loss + KL-regularized fine-tuning), and (ii) inference-time weighted resampling (Algorithm 2, Eq. 23). To isolate the contribution of each, we apply the same resampling procedure to TR2-D2 (Table 6). Two findings emerge. First, even *without* resampling, TD3B's training-time changes alone deliver a $26.5\%$ improvement in gated reward over TR2-D2 ($4.25$ vs. $3.36$, $p=0.0078$, Welch's $t$-test). Second, weighted resampling provides a complementary, additional gain on top of training-time gains ($+25.4\%$ on TD3B; both methods benefit). TD3B remains best under both conditions, indicating the components are complementary rather than redundant.

*Table 6.* Decomposing training-time vs. inference-time contributions. TR2-D2 evaluated with and without the same weighted resampling procedure used by TD3B (3-seed evaluation).

| Setting | $\mathrm{DA}(d^\star = +1)$ | $\mathrm{DA}(d^\star = -1)$ | Gated |
|---|---|---|---|
| TR2-D2 w/o resampling | $0.178 \pm 0.127$ | $0.875 \pm 0.053$ | $3.36 \pm 0.24$ |
| TR2-D2 w/ resampling | $0.508 \pm 0.011$ | $1.000 \pm 0.000$ | $4.64 \pm 0.07$ |
| TD3B w/o resampling | $0.650 \pm 0.163$ | $0.683 \pm 0.040$ | $4.25 \pm 0.20$ |
| TD3B w/ resampling | $\mathbf{0.795 \pm 0.017}$ | $\mathbf{1.000 \pm 0.000}$ | $\mathbf{5.33 \pm 0.03}$ |

# D   Algorithms

---

**Algorithm 1 Direction-Only Amortized Fine-Tuning of an MDLM**

---

1: **Input:** pre-trained MDLM $p_{\theta_0}(y)$
2:       directional dataset $\mathcal{D} = \{(y^{(n)}, a^{(n)})\}_{n=1}^N$
3:       direction oracle $f_\phi(y)$
4:       replay buffer $\mathcal{B} \leftarrow \emptyset$
5: **Hyperparameters:** learning rate $\eta$, temperature $\alpha$, contrastive weight $\lambda_{\mathrm{ctr}}$, KL weight $\lambda_{\mathrm{reg}}$
6: **while** not converged **do**
7:  |   Sample minibatch $\{(y, a)\}$ from $\mathcal{D}$
8:  |   Compute direction labels $d(y) \in \{+1, -1\}$ and weights $\kappa(y)$
9:  |   Update direction oracle parameters $\phi$ using $\mathcal{L}_{\mathrm{dir}}$
10: |                                       ▷ *Populate replay buffer with direction-aligned samples*
11: |   Sample candidate sequences $\{\tilde{y}_k\}_{k=1}^M$ from $p_\theta(y)$
12: |   **for** each $\tilde{y}_k$ **do**
13: |    |   Compute direction score $S(\tilde{y}_k) = \sigma(d^\star \cdot f_\phi(\tilde{y}_k)/\tau)$
14: |    |   Set importance weight $w(\tilde{y}_k) \propto \exp(S(\tilde{y}_k)/\alpha)$
15: |   **end for**
16: |   Add weighted samples $\{(\tilde{y}_k, w(\tilde{y}_k))\}$ to replay buffer $\mathcal{B}$
17: |                                       ▷ *Fine-tune MDLM using WDCE and contrastive objectives*
18: |   Compute $\mathcal{L}_{\mathrm{WDCE}}(\theta)$ using samples from $\mathcal{B}$
19: |   Compute $\mathcal{L}_{\mathrm{ctr}}(\theta)$ on non-negative samples
20: |   Compute regularization $\mathcal{L}_{\mathrm{reg}}(\theta) = \mathrm{KL}(p_\theta \| p_{\theta_0})$
21: |   Update $\theta \leftarrow \theta - \eta \nabla_\theta \big(\mathcal{L}_{\mathrm{WDCE}} + \lambda_{\mathrm{ctr}}\mathcal{L}_{\mathrm{ctr}} + \lambda_{\mathrm{reg}}\mathcal{L}_{\mathrm{reg}}\big)$
22: **end while**
23: **return** fine-tuned generator $p_{\theta^\star}(y)$ and oracle $f_{\phi^\star}$

---

---
**Algorithm 2 Sampling Directional Allosteric Binders**

---

1: **Input:** fine-tuned MDLM $p_{\theta^\star}(y)$
2:      desired direction $d^\star \in \{+1, -1\}$
3:      direction oracle $f_{\phi^\star}(y)$
4:      number of candidates $M$
5: Sample candidate binders $\{y_m\}_{m=1}^M$ i.i.d. from $p_{\theta^\star}(y)$
6: **for** $m = 1$ **to** $M$ **do**
7:      Compute direction score $S(y_m) = d^\star \cdot f_{\phi^\star}(y_m)$
8:      Set weight $w_m \propto \exp(S(y_m))$
9: **end for**
10: Resample $y \sim \mathrm{Categorical}(\{w_m\}_{m=1}^M)$
11: **return** $y$                                         ▷ *directionally biased binder*

---

