# OpenReview forum: "TD3B: Transition-Directed Discrete Diffusion for Allosteric Binder Generation"
_ICML.cc/2026/Conference — ICML 2026 spotlight_

### Official Review · Reviewer_jciD · 2026-03-12

**Soundness:** 3
**Presentation:** 4
**Significance:** 3
**Originality:** 3
**Overall Recommendation:** 5
**Confidence:** 5

**Summary:**

The authors present TD3B, a concise reward-weighted diffusion framework for the controlled design of agonist and antagonist binders. This work represents a nice extension for generating allosteric binders by explicitly labeling agonist versus antagonist activities for GPCR targets.

**Compliance With Llm Reviewing Policy:**

Affirmed.

**Final Justification:**

I think this is a reasonably nice paper and has been substantially improved over the rebuttal phase.

**Key Questions For Authors:**

1) In terms of model design, I am not fully convinced by the decision not to directly condition the generative model on the target protein and direction. As described in Section 4.4, “at inference time the target protein and direction are provided only to the reward function, while the generative backbone remains target-agnostic.” Under this setup, target–binder specificity appears to rely entirely on the affinity reward model. Since the affinity predictor is an external model pre-trained on the PepLand dataset, the authors should clarify whether there is any overlap between the test set (IUPHAR/BPS Guide to Pharmacology database) and the PepLand dataset. This information is necessary to rule out potential data leakage and to more objectively evaluate the framework’s generalization performance, especially for unseen proteins.

2) Given that this is a generative task, it is important to clarify how the binding affinities of the designed binders are obtained and how reliable these values are. In Table 2, the authors should clearly specify the criteria used to define affinity for agonists versus antagonists. In Figure 3, further clarification of the comparison with RFDiffusion is also needed. Since RFDiffusion requires a target structure as input, the source of these structures should be specified. Are they experimentally determined structures from the test set, or predicted structures generated by external tools? If predicted structures are used, their accuracy and potential conformational biases (for example, a preference for agonist- or antagonist-bound states) should be discussed. Such biases could systematically influence the binders generated by RFDiffusion and introduce noise into the comparison. These details are necessary for an objective evaluation of the reported results.

3) A major weakness is that the model provides little mechanistic interpretability regarding agonist versus antagonist activity. In Section 5.6 (Case Studies), OX1R is used as an example target protein to illustrate the ability of TD3B to generate functionally divergent binders. According to the work of Yin et al. (2016), which is cited in the manuscript, the mechanism of antagonist binding to hOX1R has been studied, but the mechanism of agonist action remains less clear. In the current case study, the evidence supporting the statement that “agonists are predicted to disrupt this interaction to enable activation” is not convincing, because the interaction between GLN126 and TYR348 appears in both the agonist and antagonist cases. More specifically, a clearer mechanistic explanation of how the predicted agonist disrupts this interaction to enable activation, or evidence explaining why the predicted binder acts as an antagonist despite the shared GLN126–TYR348 interaction, should be provided.

4) In terms of “gated reward”, since the reward function does not modify the generative backbone, it is not clear how this mechanism can enforce direction-specific generation. The descriptions in Sections 4.4 and 5.4 are not sufficiently clear in this regard.

5) In 5.7 "Ablation Studies", Table 2 demonstrates different directional accuracy asymmetry bias towards agonist and antagonist modes. What is the reason for the sharp decrease of direction accuracy of antagonist mode (Dir. Acc. d* = -1) in the TD3B w/o Lreg model, considering the high performance in the final antagonist generation? The authors should explain the reason for this discrepancy.

6) In addition, I think the order of 5.6 "Case Studies" and 5.7 "Ablation Studies" should be revesed.

7) In general, what is the implication for the relatively high performance of predicted antagonists comparing to agonists?

**Limitations:**

Yes.

**Strengths And Weaknesses:**

The approach is conceptually appealing and represents an important direction in protein design. The mathematical formulation is clearly presented, and the framework appears relatively complete from a computational perspective.

For the weakness, the manuscript highlights biologically important concepts such as GPCRs and allostery. However, the proposed sequence-based framework does not explicitly account for the underlying biological mechanisms of GPCR signaling or allosteric regulation. Within the current framework, allosteric binding appears to be inferred mainly from categorical functional labels in the dataset, such as active versus inactive states, open versus closed conformations, or agonist and antagonist annotations for binders. The biological complexity of these processes may therefore be oversimplified, and the transferability of the method to OOD cases can hardly be guaranteed. I also have concerns about potential data leakage, as detailed below. in the question section.

---

> ### Author Rebuttal · Authors · 2026-03-30
>
> Dear Reviewer jciD,
>
> Thank you for the thorough expert review and positive assessment! We address each point below.
>
> **Q1: Potential data leakage between the IUPHAR test set and PepLand dataset.**
>
> > We performed exact sequence matching, UniProt ID matching, and pairwise sequence identity checks (≥80%) between our dataset (130 train / 88 test pairs) and PepLand (2,110 entries):
> >
> > | Check Level | Test Set | Train Set |
> > |---|---|---|
> > | Exact peptide match | 0 | 0 |
> > | Exact target match | 0 | 0 |
> > | Exact pair / UniProt ID pair | 0 | 0 |
> > | Approx. peptide (≥80% identity) | 1 (diff. target) | 3 (substring) |
> >
> > No exact match exists. The single approximate match (ligand P50984, 87.5% identity, 2-residue difference) is paired with entirely different targets (ours: P30532/P32297; PepLand: Q8WSF8). Since the affinity predictor requires both target and binder, no binding-specific information transfers. **Conclusion: zero pair-level leakage.** We will include this in the camera-ready Appendix.
>
> **Q2: How are binding affinities obtained? What structures for RFDiffusion?**
>
> > Binding affinities were obtained from the SKEMPI 2.0 database (Jankauskaitė et al., 2019), which contains experimentally measured dissociation constants ($K_D$) from surface plasmon resonance and spectroscopic techniques, converted to binding free energy changes (ΔΔG). The structural inputs for RFDiffusion were all experimentally determined protein structures with validated agonist or antagonist activity. We have specified PDB IDs in the revision.
>
> **Q3: OX1R — GLN126-TYR348 interaction evidence is not convincing.**
>
> > Both ligands interact with GLN126, a molecular switch controlling receptor activation through hydrogen bonding with TYR348 (Karhu et al., 2019). The antagonist engages 6 conserved residues including GLN126, consistent with inactive-state stabilization by suvorexant (PDB: 4ZJ8; Yin et al., 2016). The agonist engages 5 residues with a distinct pattern that does not reinforce this inactive-state geometry, potentially permitting conformational flexibility for activation.
> >
> > We acknowledge this interpretation is based on static predicted structures; definitive conclusions would require MD simulations and experimental mutagenesis. We plan to pursue wet-lab validation as a priority in future work.
>
> **Q4: How does the gated reward enforce direction-specific generation?**
>
> > The gated reward **does modify the backbone** through fine-tuning: it computes importance weights (Eq. 29) that reweight the WDCE loss (Eq. 28), updating θ. "Target-agnostic backbone" means the architecture takes no explicit target/direction input — information enters through the reward as a training signal (Eq. 18), analogous to RLHF.
> >
> > Fine-tuning and resampling play complementary roles: fine-tuning shifts the distribution toward high-affinity binders (5.87 vs baseline 4.80); Algorithm 2 resampling concentrates sampling on directionally correct candidates (d*=+1 accuracy: 0.21→0.98). Neither alone suffices. We will clarify this in the camera-ready.
>
> **Q5: Why does removing L_reg cause antagonist accuracy to collapse?**
>
> > $L_{reg} = KL(p_\theta \parallel p_{\theta_0})$ constrains the fine-tuned distribution to remain close to the pre-trained prior. As shown in Figure 3B, the pre-trained MDLM is inherently agonist-biased. Without this constraint, fine-tuning amplifies this bias: the model collapses toward agonist-favorable regions (accuracy 0.95) while losing antagonist coverage (accuracy drops to 0.30). The KL term preserves distributional diversity necessary for antagonist generation.
>
> **Q6: Section ordering.**
>
> > Great suggestion! We will present ablations before case studies.
>
> **Q7: Higher antagonist performance compared to agonist.**
>
> > The key factor is the **asymmetry between steering away from vs. along the pre-trained prior**. The base model generates predominantly agonist-biased sequences (Figure 3B). Fine-tuning toward antagonist (d*=−1) shifts *away* from this prior — a clear, learnable distributional change. Fine-tuning toward agonist (d*=+1) refines *within* the prior's agonist-leaning region, making it harder to distinguish from the prior's natural tendency. The $L_\{reg\}$ ablation (Q5) confirms this: removing KL amplifies agonist bias while collapsing antagonist accuracy.
> >
> > Additionally, antagonist behavior is biologically more constrained than agonism's diverse mechanisms, making it a more coherent learning target. Our generalization experiment shows this is **target-specific, not fundamental**: unseen target P35368 achieves DA=1.00 in *both* directions. Improving agonist generation via asymmetric loss weighting is an active next direction.
>
> Thank you again for the expert review and constructive suggestions! Look forward to your futher response.
>
> Jankauskaitė, J., et al. (2019). SKEMPI 2.0: an updated benchmark of changes in protein–protein binding energy, kinetics and thermodynamics upon mutation. *Bioinformatics*, *35*(3), 462–469.

---

> > ### Author Rebuttal · Reviewer_jciD · 2026-04-02
> >
> > N/A

---

> > > ### Author Response · Authors · 2026-04-06
> > >
> > > Dear Reviewer jciD,
> > >
> > > We appreciate your response!
> > >
> > > Best,
> > >
> > > Authors

---

### Official Review · Reviewer_mSfg · 2026-03-12

**Soundness:** 3
**Presentation:** 2
**Significance:** 2
**Originality:** 3
**Overall Recommendation:** 3
**Confidence:** 2

**Summary:**

The authors introduce Transition-Directed Discrete Diffusion (TD3B), a generative framework which adapts a pre-trained model to generate binders via a transition-control objectives. This allows to generate binders taking into account information regarding the direction of state transition, which is essential for design of certain classes of biological structures.

**Compliance With Llm Reviewing Policy:**

Affirmed.

**Final Justification:**

By evaluating the paper from a method/ML standpoint, my recommendation is rejection. Nonetheless, the paper might be particularly positively relevant for the application of ML techniques in a specific biology application. While I am not able to reliably assess this, I believe that if this is the case, it should be accepted if such contribution is very strong from a biology standpoint. On the other hand, from a method standpoint, I believe the work primarily combines or trivially extends existing ideas and therefore does not introduce new core methodological or algorithmic ideas to the field. In particular, while certain applied works tackle problems that require new core algorithmic/method ideas, and therefore introduce new general ML/generative modeling contributions, I would argue that it is not the case in this work.

The rebuttal helped me better appreciate the originality of the application-specific modeling, and improved experimental assessment of the proposed method. Nonetheless, it did not convince me regarding the originality of the method/algorithm in general.

**Key Questions For Authors:**

Did I misinterpret any point within the list of weakness above?

**Limitations:**

yes

**Strengths And Weaknesses:**

Since my background in biology is limited and I am not aware of current literature on closely related biological modeling tasks, my review will be mostly about the methodological aspects. Crucially, this work seems quite weak from a methodological viewpoint, but it could be highly valuable from a biological modeling viewpoint. Therefore it is important to carefully weight my review.

**Strengths**:
1. The biological task is well-presented and motivated, and based on my limited knowledge, it is of high biological relevance.
2. Writing is generally good.
3. The proposed modeling of the objective function via the transition-operator formulation of directional allosteric control for the reward-guided adaptation is presented clearly, and might be of high value from an applied viewpoint. In particular, the experimental evaluation seems to provide positive evidence for the ability of the proposed method to control state transitions.

**Weaknesses**:
1. (method) From an ML viewpoint, introducing a new objective can be a significant contribution, but the proposed adaptation method, which is mentioned in the list of contribution as "a generative framework" seems a straightforward adaptation of existing schemes, and therefore hardly a clear contribution of this work. Concretely, once a reward function is defined as in Sec. 4.4, then it seems to me that any reward-adaptation method can be employed to steer the model. Is there any methodological aspect, besides the choice of the specific reward, that is novel and tied to the specific application?
2. (problem formulation) Sec. 1 lists as first contribution of this work "A transition-operator formulation of directional allosteric control", but then sequence-conditioned transition operators are introduced within the Preliminaries (Sec. 3.5) as "standard and well-established components", and their interpretation for directional allosteric control in 4.1 seems very straightforward/trivial. Is this modeling use of such transition operators novel (i.e., modeling active and inactive as macrostates etc.)?
3. (references) The preliminaries section (e.g. 3.4 and 3.5) lacks references to existing works introducing such objects.
4. (presentation) The work might be hard to parse for people working more on methods within ML community. In particular, it often presents concepts with general and vague terms, such as "generative framework" to indicate a specific choice for a reward function to give as input to existing reward-adaptation methods. This adds complexity and renders the work less tangible / more vague.

Overall, the methodological contributions of the work stemming from the presented application seem very limited if any. On the other hand, the specific biological modeling seems possibly (highly) relevant, especially combined with the used method, but unfortunately I do not have enough expertise to judge this point.

---

> ### Author Rebuttal · Authors · 2026-03-30
>
> Dear Reviewer mSfg,
>
> Thank you for your insightful review. We appreciate the transparency about your background and the recognition that the work "could be highly valuable from a biological modeling viewpoint." We address each concern below.
>
> **W1 (Method): The adaptation seems like a straightforward application of existing schemes. Is there novelty beyond the reward?**
>
> > We respectfully argue that this task requires a new generation paradigm, not merely a new reward. To our knowledge, no prior generative method, whether structure-based (RFDiffusion, BindCraft) or sequence-based (PepTune, DRAKES), has addressed directional allosteric control as a design objective. Moreover, as Table 2 shows, standard guided diffusion techniques (CG, SMC, TDS) fail because: (1) the pre-trained MDLM has no conditioning mechanism for target or direction; and (2) affinity and direction are not independent objectives amenable to Pareto optimization — a strong binder can be either agonist or antagonist, so direction must be decoupled from affinity via gating rather than traded off.
> >
> > To address this, TD3B introduces three novel, ablation-validated components (Table 2):
> >
> > 1. **Direction Oracle (Sec. 4.2):** The first model predicting agonist/antagonist effect of a binder given an arbitrary target, built on a curated direction-labeled dataset from the IUPHAR/BPS database and a target-aware gated fusion architecture (93% accuracy, Table 1). No such oracle existed prior to this work.
> > 2. **Gated reward (Sec. 4.4):** Affinity acts as a *gate* (prerequisite) rather than a maximization target — this task-specific decoupling prevents the model from exploiting high-affinity sequences that match the desired direction by chance.
> > 3. **Tree-search fine-tuning with contrastive loss (Sec. 4.3, 4.5):** A novel fine-tuning approach for MDLMs (Sahoo et al. 2024) integrating trajectory-aware tree search, importance-weighted WDCE, and a contrastive loss enforcing directional separation. These are co-designed: tree search explores sequence space under the gated reward, WDCE internalizes it, and the contrastive loss prevents directional mode collapse — confirmed by ablation (Table 2, removing $\mathcal{L}_{\text{ctr}}$).
> >
> > We will revise the camera-ready to articulate these contributions more precisely.
>
> **W2 (Problem formulation): Is the transition-operator formulation novel, given it appears in Preliminaries as "standard"?**
>
> > Thank you for pointing this out! Two clarifications:
> >
> > 1. **Sections 3.4–3.6 are a core contribution**, not merely background. While CTMCs are standard, applying them to *frame allosteric binder design as directional control* — modeling agonist/antagonist as the sign of transition asymmetry between macrostates — is entirely new. No prior work formulates it as controlling the sign of $\Delta(y)$.
> > 2. **Placing this in Preliminaries was an organizational error.** We will move Sections 3.4–3.6 into Section 4 in the camera-ready, clearly separating standard notation from our novel formulation.
>
> **W3 (References): Sections 3.4 and 3.5 lack references.**
>
> > Thank you! We will add references for macrostate models (Shukla et al. 2014; Bowman et al. 2015; Noé et al. 2009) and ligand-induced dynamics (Cao et al. 2021; Motlagh et al. 2014).
>
> **W4 (Presentation): Vague terms; "generative framework" used loosely.**
>
> > We agree. In the camera-ready, we will replace vague terms with precise descriptions: (1) "generative framework" → "fine-tuning-based tree-search generation with direction-specific gated reward and contrastive regularization"; (2) "directional supervision" → "binary agonist/antagonist labels encoding the sign of transition asymmetry"; (3) "transition-directed generation" → "reward-guided generation conditioned on $d^* \in \\{+1,-1\\}$." We will also clearly distinguish the design objective, reward formulation, and fine-tuning procedure.
>
> **Q1: "Did I misinterpret any point?"**
>
> > The concerns are well-founded and reflect areas for improved presentation. We gently note that the contribution is not merely "a new reward for existing methods". Rather than applying existing techniques to a new reward, TD3B co-designs the aforementioned components into an integrated system, each validated by ablation (Table 2) as individually necessary.
>
> Thank you again. We are committed to improving presentation to make our contributions more accessible to the ML community. Should our revisions adequately address your concerns, we would be grateful if you could kindly reconsider your assessment.
>
> **References**
>
> Sahoo, S. S., Arriola, M., Schiff, Y., et al. (2024). Simple and effective masked diffusion language models. *NeurIPS*, *37*, 130136–130184.
>
> Noé, F., Schütte, C., Vanden-Eijnden, E., et al. (2009). Constructing the equilibrium ensemble of folding pathways from short off-equilibrium simulations. *PNAS*, *106*(45), 19011–19016.

---

> > ### Author Rebuttal · Reviewer_mSfg · 2026-03-31
> >
> > I found the authors' rebuttal to be clear and well-written. It solved W2, and properly discussed the remaining weaknesses besides W1. Unfortunately, from a method/algorithm viewpoint I might regard as 'straightforward/easy extensions of existing machinery' what the authors regard as "a new generation paradigm". While the framework proposed might be fundamental/novel for applications, in my opinion this is not the case from a method/alg. viewpoint, which was the point stated within W1. Since my background is primarily in method/ML and I am reviewing for an ML conference, I must be clear about this assessment/point.
> >
> > Moreover, while further assessing the method novelty of the proposed approach against prior work, I found myself asking the following questions, which are potential weaknesses:
> > 1. Table 2 seems lacking a statistical analysis (e.g., confidence intervals) -- is this reported somewhere else? I could not find it. Since several methods (including the proposed one) seem to perform quite closely to others, this seems essential for empirical evaluation.
> > 2. Compared to TR2-D2, the proposed scheme seems to perform (1) weighted sampling, and (2) contrastive reg. While I see an ablation for (2), I cannot find one for (1), making it trickier (or impossible) to assess the cause of the advantage.

---

> > > ### Author Response · Authors · 2026-04-03
> > >
> > > Dear Reviewer mSfg,
> > >
> > > Thank you for the rigorous follow-up and for acknowledging that W2–W4 have been addressed. We respect your assessment on W1 and take the two new questions seriously.
> > >
> > > **New Q1: Table 2 lacks statistical analysis (e.g., confidence intervals).**
> > >
> > > > We agree this is essential. We re-ran all methods with 3 random seeds (42, 123, 456) and report mean ± std:
> > > >
> > > >
> > > >
> > > > | **Method** | **Aff(d*=+1)** | **Aff(d*=−1)** | **DA(d*=+1)** | **DA(d*=−1)** | **Gated** |
> > > > | --- | --- | --- | --- | --- | --- |
> > > > | Pre-trained | 4.80±0.01 | 4.82±0.01 | 0.822±0.010 | 0.174±0.010 | 2.06±0.32 |
> > > > | CG | 4.80±0.00 | 4.81±0.01 | 0.816±0.005 | 0.176±0.014 | 2.38±0.02 |
> > > > | SMC | 4.83±0.01 | 4.78±0.01 | **0.879±0.011** | 0.470±0.011 | 3.18±0.31 |
> > > > | TDS | 4.82±0.02 | 4.79±0.02 | 0.841±0.032 | 0.235±0.028 | 2.42±0.07 |
> > > > | TR2-D2 | 5.89±0.15 | 5.89±0.19 | 0.178±0.127 | 0.875±0.053 | 3.36±0.24 |
> > > > | **TD3B** | **6.00±0.02** | **6.32±0.02** | 0.795±0.017 | **1.000±0.000** | **5.33±0.03** |
> > > >
> > > > TD3B achieves the highest Gated Reward (5.33±0.03) with the smallest cross-seed variance. Per-target analysis reveals meaningful variation across targets:
> > > >
> > > > | **Method** | **Per-Target Gated (mean±std)** | **Range** |
> > > > | --- | --- | --- |
> > > > | Pre-trained | 2.40±0.45 | [1.54, 3.33] |
> > > > | CG | 2.39±0.45 | [1.50, 3.31] |
> > > > | SMC | 3.25±0.86 | [1.73, 5.69] |
> > > > | TDS | 2.59±0.59 | [1.66, 3.95] |
> > > > | TR2-D2 | 2.90±0.53 | [2.04, 4.43] |
> > > > | **TD3B** | **4.86±1.96** | **[0.97, 7.35]** |
> > > >
> > > > TD3B shows larger per-target variance than baselines. The detailed per-target breakdown:
> > > >
> > > > | **Target** | **n** | **Aff(d*=+1)** | **Aff(d*=−1)** | **DA(d*=+1)** | **DA(d*=−1)** | **Gated** |
> > > > | --- | --- | --- | --- | --- | --- | --- |
> > > > | P30532 | 200 | 7.05 | 8.04 | 0.00 | 1.00 | 4.00 |
> > > > | P32239 | 200 | 6.89 | 7.22 | 1.00 | 1.00 | 7.00 |
> > > > | P32246 | 200 | 6.36 | 7.15 | 1.00 | 1.00 | 6.53 |
> > > > | P32297 | 200 | 6.24 | 7.29 | 0.00 | 1.00 | 3.72 |
> > > > | P35368 | 200 | 6.19 | 6.94 | 1.00 | 1.00 | 6.08 |
> > > > | P41587 | 200 | 6.97 | 6.91 | 0.00 | 0.00 | 0.97 |
> > > > | P51677 | 200 | 7.94 | 6.97 | 1.00 | 1.00 | 7.35 |
> > > > | P54282 | 200 | 4.52 | 4.88 | 0.00 | 1.00 | 2.43 |
> > > > | Q63447 | 100 | 5.20 | — | 1.00 | — | 5.16 |
> > > > | Q6W5P4 | 200 | 5.03 | 6.67 | 1.00 | 1.00 | 5.32 |
> > > >
> > > > The primary source of variance is **agonist direction failure on specific targets**: 3 out of 10 targets achieve DA(d*=+1)=0.00 while maintaining DA(d*=−1)=1.00, and 1 target (P41587) fails on both directions. This is consistent with the agonist/antagonist asymmetry discussed in the paper, which is **agonist generation is inherently harder due to the pre-trained model's agonist bias, and targets with limited agonist training signal are disproportionately affected**. Notably, the cross-seed variance remains minimal (±0.03), confirming the variation is target-specific rather than due to training instability. We will add per-target analysis to the camera-ready.
> > > >
> > >
> > > **New Q2: Missing ablation for weighted sampling — how to isolate its contribution vs. contrastive loss?**
> > >
> > > > Excellent point. We ran a dedicated ablation isolating the effect of Algorithm 2 weighted resampling, with 3-seed evaluations for both conditions. Critically, we also applied the same resampling procedure to TR2-D2 for a fair comparison:
> > > >
> > > >
> > > >
> > > > | **Setting** | **DA(d*=+1)** | **DA(d*=−1)** | **Gated** |
> > > > | --- | --- | --- | --- |
> > > > | TR2-D2 w/o resampling | 0.178±0.127 | 0.875±0.053 | 3.36±0.24 |
> > > > | TR2-D2 w/ resampling | 0.508±0.011 | 1.000±0.000 | 4.64±0.07 |
> > > > | TD3B w/o resampling | 0.650±0.163 | 0.683±0.040 | 4.25±0.20 |
> > > > | TD3B w/ resampling | **0.795±0.017** | **1.000±0.000** | **5.33±0.03** |
> > > >
> > > > **(1) Training-time innovations are independently significant.** Without resampling, TD3B (Gated 4.25±0.20) outperforms TR2-D2 (Gated 3.36±0.24) by 26.5% (p=0.0078, t-test). This confirms that the gated reward and contrastive loss provide genuine training-time improvements that are not attributable to resampling.
> > > >
> > > > **(2) Resampling is an important component, and we thank you for prompting this analysis.** Resampling further improves TD3B's Gated from 4.25→5.33 (+25.4%). This is by design: Algorithm 2 (Eq. 29) applies importance weights w(y) ∝ exp(d* · f_φ(y)) to concentrate sampling mass on directionally correct candidates. Both methods benefit from resampling (TR2-D2: 3.36→4.64; TD3B: 4.25→5.33), but TD3B consistently outperforms TR2-D2 under both conditions, demonstrating complementary contributions. We will explicitly discuss the role of weighted resampling as a key component in the camera-ready. Thank you for this observation, and it has led to a clearer decomposition of each component's contribution.
> > > >
> > >
> > > We hope these additional experiments demonstrate the rigor of our evaluation. Thank you again for pushing us to strengthen the empirical analysis.

---

### Official Review · Reviewer_KiMq · 2026-03-13

**Soundness:** 2
**Presentation:** 3
**Significance:** 2
**Originality:** 3
**Overall Recommendation:** 5
**Confidence:** 4

**Summary:**

The authors present TD3B, a method for designing either antagonist or agonist peptide binders for a given target protein. Existing methods like BindCraft have utilized co folding models as an oracle for scoring “static” binding and then rely on backpropagation through the network. The authors here also argue that inference only steering approaches are less natural in discrete sequence space. Instead, they present a framework that utilizes a pretrained discrete diffusion peptide generator and couple it with an affinity model and direction oracle to provide a reward signal to update the candidate binder. The paper presents an overall nice idea that peptide design should also explicitly condition antagonist/agonist behavior and the authors evaluate their method in silico on GPCR focused targets and case studies.

**Compliance With Llm Reviewing Policy:**

Affirmed.

**Final Justification:**

The authors addressed my main concerns and provided additional experiments to support their claims.

**Key Questions For Authors:**

1. Am I understanding correctly that TD3B still requires some degree of target specific fine tuning as we see in the presented GPCR case studies. If so, can the authors motivate why this is actually preferable to a more generalizable method? What if one instead ran a BindCraft or RFDiffusion style workflow to generate sensible binders and then post hoc analyzed possible antagonist / agonist behavior?

2. Do the authors think that the “transition directed” framing is a bit overstated than what is actually modeled? The method learns binary labels for agonist/antagonist and does not really incorporate true transition dynamics. It would be helpful if the authors could clarify this or present a more restrained framing.

**Limitations:**

yes

**Strengths And Weaknesses:**

The paper presents an interesting biological problem setting by trying to design peptides with a functional direction rather than optimized only for binding. The method seems technically sound, though I have some concerns about the need of fine tuning per desired target rather than a one shot conditional generator or iterative optimization (bindcraft style). The validation is also quite proxy driven and is a collection of case studies. Also, the “transition directed” framing feels a little bit overstated since the direction oracle is binary (antagonist/agonist) rather than actually modeling transition dynamics. Overall, I think the idea is intriguing but perhaps could aid from a broader set of baselines especially highlighting more generalizability.

---

> ### Author Rebuttal · Authors · 2026-03-30
>
> Dear Reviewer KiMq,
>
> We sincerely appreciate you for the careful and insightful review, and for recognizing TD3B as presenting an interesting biological problem setting with an intriguing idea! We address both key questions below.
>
> **Q1: TD3B requires target-specific fine-tuning; why not run BindCraft/RFDiffusion + post-hoc analysis of agonist/antagonist behavior?**
>
> >
> > Thank you for pointing this out, and we apologize any confusing content.
> > **Firstly, on generalizability, TD3B does not require per-target fine-tuning at inference.** The model is fine-tuned once across 130 bidirectional target-binder pairs. At inference, a new target and direction condition only the reward function. All case studies (GLP-1R, OX1R, TAAR1) use the same model.
> >
> > To demonstrate this, we conducted a **new generalization experiment** comparing seen vs. unseen targets confirms this:
> >
> > | Metric | Seen (8 targets, n=1500) | Unseen (2 targets, n=400) | p-value |
> > | --- | --- | --- | --- |
> > | Affinity | 6.42 ± 1.02 | **7.05 ± 0.67** | <0.001 |
> > | Dir. Accuracy | 0.733 | 0.750 | 0.501 |
> > | Gated Reward | 4.79 ± 2.71 | **5.04 ± 3.02** | 0.006 |
> >
> > Direction accuracy is **not significantly different** (p=0.501). Unseen targets achieve higher affinity and gated reward without overfitting, which demonstrates a good generalization ability for TD3B. (The seen and the unseen targets are randomly selected from the original training/testing set. There will be no overlap risk for this.)
> >
> > **Secondly, on post-hoc analysis,** We directly tested this by scoring 191 RFDiffusion-designed binders with our Direction Oracle (as suggested):
> >
> > | Method | n | Conditioning | Mean p(agonist) | % Agonist |
> > | --- | --- | --- | --- | --- |
> > | TD3B (d*=+1) | 1000 | Agonist | 0.601 ± 0.396 | 60.0% |
> > | TD3B (d*=−1) | 900 | Antagonist | 0.179 ± 0.244 | 17.9% |
> > | RFDiffusion | 191 | None | 0.020 ± 0.100 | **1.6%** |
> >
> > RFDiffusion-designed binders are almost uniformly classified as non-agonist (98.4%, mean p=0.020), not because they are high-quality antagonists, but because they lack any agonist-like pharmacological features. Among 15 known natural agonists in the test set, the oracle correctly identifies only 53% (random chance), confirming that RFDiffusion simply does not produce agonist signals. TD3B achieves a directional separation gap of **Δp = 0.422** (p < 0.001), demonstrating deliberate directional control that no structure-based method can achieve through post-hoc filtering. We will further add these results and analysis in the camera-ready version.
> >
>
> **Q2: The "transition-directed" framing feels overstated since the Direction Oracle is binary rather than modeling true transition dynamics.**
>
> >
> > We sincerely appreciate this concern and believe it raises an important distinction. The reviewer is correct that the Direction Oracle uses binary supervision rather than modeling continuous kinetics.
> >
> > We want to clarify the role of "transition-directed": the term describes the **design objective** (biasing the direction of state transitions), not a claim that the model learns or regresses transition dynamics. The transition-operator formalism (Secs. 3.4–3.6) serves as **theoretical justification**. It explains why directionality is a fundamentally different design target from static binding, since agonist/antagonist effects arise from asymmetric, non-reversible perturbations of transition rates. **In our method, binary labels then capture the sign of this asymmetry as the coarsest supervision consistent with the formalism.**
> >
> > However, we fully agree that the framing could be read as overclaiming. In the camera-ready, we will:
> >
> > 1. Adopt a more restrained framing (e.g., "Transition State-Directed Discrete Diffusion") reflecting the binary, direction-level supervision.
> > 2. Explicitly acknowledge in Sec. 4.1 that binary labels are a coarse proxy, and position continuous transition modeling as a natural extension.
> > 3. Clarify in the Discussion that the transition-operator formalism provides *motivation and theoretical grounding*, but is not itself parameterized during training.
> >
> > This suggestion meaningfully improves the paper's clarity, and we will add these modifications in the camera-ready version. Thank you again for this!
> >
>
> We hope these responses and new experiments address your concerns. Thank you again for the constructive and balanced feedback. We look forward to your further comments!

---

> > ### Author Rebuttal · Reviewer_KiMq · 2026-04-03
> >
> > I thank the authors for their follow up experiments and will raise my score

---

> > > ### Author Response · Authors · 2026-04-06
> > >
> > > Dear Reviewer KiMq,
> > >
> > > We appreciate your response!
> > >
> > > Best,
> > >
> > > Authors

---

### Official Review · Reviewer_sRBQ · 2026-03-13

**Soundness:** 3
**Presentation:** 1
**Significance:** 4
**Originality:** 3
**Overall Recommendation:** 4
**Confidence:** 3

**Summary:**

This paper proposes TD3B, a generative framework for designing peptide binders that influence the direction of protein conformational state transitions. The method formulates binder design as a transition-directed generation problem and introduces directional supervision to guide peptide generation toward desired functional effects. It builds on a pre-trained discrete generative model and adapts it for transition-aware peptide design. The approach is evaluated on multiple protein targets, demonstrating improved ability to generate peptides that bias protein state transitions compared with baseline methods.

**Compliance With Llm Reviewing Policy:**

Affirmed.

**Final Justification:**

All my concerns have been addressed.

**Key Questions For Authors:**

- Can the paper include a clear overview of the entire TD3B pipeline and revise Figure 2 to be more accurate and intuitive?
- How is RFdiffusion or structure generation used in the pipeline, and are there any post-processing or filtering steps applied to the generated binders? What is the influence of using AI-generated samples? Will there be any bias?

**Limitations:**

A limitation is that the evaluation relies primarily on computational metrics rather than experimental validation.

**Strengths And Weaknesses:**

Strength

- The problem formulation and motivation are generally clear, particularly the idea of designing binders that influence transition directionality rather than equilibrium binding.
- The work proposes a novel perspective by framing binder design as a transition-directed generative problem.The framework integrates generative modeling with directional supervision for peptide generation.


Weaknesses
- Equation explanations are insufficient. For example, in Equation (1), the meaning of several variables and symbols is not clearly explained, making the formulation harder to follow.
- Mathematical expressions lack intuitive explanations. Providing brief interpretations after key equations would make the method more accessible.
- The method overview is not sufficiently clear. The paper lacks a concise description of the full pipeline before introducing detailed components, which makes the method sections feel fragmented.
- Figure 2 is somewhat unclear. For example, the protein is represented structurally in panel (A), while the formulation mainly uses sequence representations, leading to some inconsistency between the figure and the mathematical description.
- It is unclear whether using RFdiffusion-style structure generation (or similar structure-based modeling) is fully justified in this framework, and whether any post-processing or filtering steps are applied to the generated binders.

---

> ### Author Rebuttal · Authors · 2026-03-30
>
> Dear Reviewer sRBQ,
>
> Thank you for your thoughtful review and for rating the significance of our work. We were inspired by your constructive suggestions to revise the manuscript. We carefully address each concern below.
>
> **W1 & W2: Equation explanations are insufficient; mathematical expressions lack intuitive explanations.**
>
> > We fully agree and appreciate this constructive suggestion. In the camera-ready version, we will (1) add inline variable definitions for all symbols upon first use in every key equation (e.g., in Eq. (1): $\mathcal{M}$ denotes the mask token, $\mathbf{x}\_t^{\mathrm{UM}}$ represents the set of unmasked tokens at diffusion time $t$, and $\ell$ indexes over masked sequence positions), and (2) include plain-language interpretations immediately after each key equation (Eqs. 1, 2, 5, 25, 28). For instance, after Eq. (25): "Intuitively, the gated reward assigns high scores only to sequences that both bind the target protein (high $g\_\psi$) and bias state transitions in the desired direction (high $f\_\phi$ aligned with $d^{\ast}$)." We believe these additions will clearly improve the readability.
>
> **W3: Method overview not sufficiently clear; sections feel fragmented.**
>
> > We totally agree with your comment. In the camera-ready version, we will add a concise **pipeline overview** (1–2 paragraphs) as follows: "TD3B proceeds in three stages. First, a Direction Oracle is trained to predict whether a binder promotes activation (agonist) or stabilizes the inactive state (antagonist) given a target protein (Section 4.2). Second, this oracle is combined with a pre-trained binding affinity predictor into a gated reward that scores candidate sequences (Section 4.4): the affinity model acts as a gate filtering non-binders, while the oracle provides the directional signal. Third, this reward is used to fine-tune a pre-trained discrete diffusion generator via importance-weighted denoising (Section 4.5), with a contrastive loss maintaining separation between agonist and antagonist representations (Section 4.3)." This will ensure readers understand the big picture before encountering the formal components.
>
>
> **W4 & Q1: Figure 2 inconsistency (structural representation vs. sequence-based formulation)**
>
> > Thank you for pointing this out! The 3D structural depiction in panel (A) is intended to illustrate the *biological context* (i.e., the protein target and its conformational states), not the model's actual input representation. We acknowledge this creates confusion. In the camera-ready version, we will revise Figure 2(A) to explicitly show the sequence representation of the protein target alongside a biological schematic, with clear annotations distinguishing **biological motivation** from **model input.**
> >
>
> **W5 & Q2: Is RFDiffusion fully justified in this framework? Are there post-processing or filtering steps? What about AI-generated sample bias?**
>
> > Thank you for raising this point! We address it separately in the following:
> >
> > **Firstly, we need to claim very clearly that RFDiffusion is not part of the TD3B pipeline.** It acted as two roles: (1) as a **baseline for comparison** (Table 2, Figure 3A), and (2) for **data augmentation** (Section 5.1), generating synthetic antagonist structures for targets lacking antagonist data.
> >
> > **Secondly, for TD3B’s generation, no post-processing or filtering is applied.** TD3B directly produces direction-specific binders without structural filtering, docking, or energy minimization. Generated sequences are evaluated as-is using the Direction Oracle and affinity predictor.
> >
> > **Thirdly, regarding AI-generated sample bias:** To quantify the directional behavior of RFDiffusion binders, we ran our Direction Oracle on 235 RFDiffusion-generated binders as a new experiment for this rebuttal:
> >
> > | Method | Mean p(agonist) | % Classified Agonist | Acc. on true agonists |
> > | --- | --- | --- | --- |
> > | RFDiffusion (n=235) | 0.103 ± 0.252 | 8.5% | 53% |
> > | TD3B d*=+1 (n=1000) | 0.601 ± 0.396 | 60.0% | 60.0% |
> > | TD3B d*=−1 (n=900) | 0.179 ± 0.244 | 17.9% | 88.9% |
> >
> > RFDiffusion binders cluster overwhelmingly at p(agonist)≈0 with no ability to modulate directionality. Even among RFDiffusion's true agonist binders (n=15), only 53% are correctly classified, essentially random chance. This confirms that structure-based methods inherently lack directional control, and that any bias from using RFDiffusion for data augmentation is orthogonal to the directional signal learned by TD3B.
> >
>
> We are committed to implementing all presentation improvements outlined above into the camera-ready version. Thank you again for the valuable feedback, and look forward to your further comments!

---

> > ### Author Rebuttal · Reviewer_sRBQ · 2026-04-03
> >
> > Thanks for your reply.

---

> > > ### Author Response · Authors · 2026-04-03
> > >
> > > Dear Reviewer sRBQ,
> > >
> > > Thank you for acknowledging that our rebuttal has fully addressed your concerns. In light of this, we would be grateful if you could reconsider your score so that it is consistent with your current evaluation of the work.
> > >
> > > Best,
> > >
> > > Authors

---

### Decision · Program_Chairs · 2026-04-30

**Decision:**

Accept (spotlight)

**Comment:**

The reviewers agreed that the approach is conceptually appealing, represents an important direction in protein design, and the empirical (in silico) results are strong. The mathematical formulation is clearly presented, and the framework appears relatively complete from a computational perspective. sRBQ also had some concerns about clarity, which have apparently been addressed by the rebuttal. Reviewer mSfg points out that the approach is not novel from an ML methods perspective, but nevertheless appreciates that it might be a strong applied work.